# D-amino acid oxidase promotes cellular senescence via the production of reactive oxygen species

Taiki Nagano[1,2], Shunsuke Yamao[2], Anju Terachi[2], Hidetora Yarimizu[2], Haruki Itoh[2], Ryoko Katasho[3], Kosuke Kawai[2], Akio Nakashima[1,4], Tetsushi Iwasaki[1,2,3], Ushio Kikkawa[1,4], Shinji Kamada[1,2,3]

**D-amino acid oxidase (DAO) is a flavin adenine dinucleotide (FAD)–dependent oxidase metabolizing neutral and polar D-amino acids. Unlike L-amino acids, the amounts of D-amino acids in mammalian tissues are extremely low, and therefore, little has been investigated regarding the physiological role of DAO. We have recently identified *DAO* to be up-regulated in cellular senescence, a permanent cell cycle arrest induced by various stresses, such as persistent DNA damage and oxidative stress. Because DAO produces reactive oxygen species (ROS) as byproducts of substrate oxidation and the accumulation of ROS mediates the senescence induction, we explored the relationship between DAO and senescence. We found that inhibition of DAO impaired senescence induced by DNA damage, and ectopic expression of wild-type DAO, but not enzymatically inactive mutant, enhanced it in an ROS-dependent manner. Furthermore, addition of D-amino acids and riboflavin, a metabolic precursor of FAD, to the medium potentiated the senescence-promoting effect of DAO. These results indicate that DAO promotes senescence through the enzymatic ROS generation, and its activity is regulated by the availability of its substrate and coenzyme.**

## Introduction

D-amino acid oxidase (DAO) is a flavin adenine dinucleotide (FAD)–dependent peroxisomal enzyme that catalyzes the oxidation of neutral and polar D-amino acids with a strict stereospecificity to give $\alpha$-keto acids, ammonia, and reactive oxygen species (ROS) (Pollegioni et al, 2007). D-amino acids are usually present only in very low amounts in higher organisms, including human beings, but are increasingly recognized as physiologically functional regulators involved in several biological processes (Ohide et al, 2011). For example, D-serine acts as an agonist of NMDA receptors that are ligand-gated ion channels mediating excitatory neurotransmission in the brain, and hence, alteration of D-serine metabolism is relevant

for neurological diseases, such as schizophrenia, ischemia, epilepsy, and neurodegenerative disorders (Schell et al, 1995; Katsuki et al, 2004; Billard, 2012). Because DAO controls the D-serine concentration in the brain, the relationships between DAO and these neurological diseases have been extensively investigated (Hashimoto et al, 2005; Maekawa et al, 2005; Madeira et al, 2008; Verrall et al, 2010). However, despite the broad expression pattern of DAO in human tissues such as the kidney and liver, much less is known about its function other than in the nervous system.

Cellular senescence is defined as a persistent form of cell cycle arrest induced by various genotoxic stresses, such as DNA-damaging agents, ROS, and activation of oncogenes, and is implicated in both tumor inhibition and age-related disorders presumably through the secretion of a variety of inflammatory factors, referred to as the senescence-associated secretory phenotype (d'Adda di Fagagna, 2008; Yanai & Fraifeld, 2017; Kuilman et al, 2010; Campisi, 2013; Salama et al, 2014; Lu & Finkel, 2008; Serrano et al, 1997; Chen & Ames, 1994; Di Leonardo et al, 1994). In response to genotoxic stresses, the DNA damage response pathway activates the tumor suppressor p53, a transcription factor whose activity is required for the initiation and maintenance of senescence (Vousden & Prives, 2009; Rufini et al, 2013). The activated p53 induces a large number of genes involved in senescence, such as the cyclin-dependent kinase inhibitor *p21* and *proline dehydrogenase* (*PRODH*) (el-Deiry et al, 1993; Romanov et al, 2012; Nagano et al, 2016, 2017). Although p21 has been considered as a central mediator of p53-dependent cell cycle arrest, the full set of p53 target genes needed for senescence remains to be elucidated (Noda et al, 1994; Brady et al, 2011; Romanov et al, 2012; Valente et al, 2013). Besides the activation of p53, ROS also critically contribute to senescence (Lu & Finkel, 2008; Campisi, 2013; Salama et al, 2014). The accumulation of ROS is widely observed in senescence induced by various types of stress. ROS can hasten senescence through induction of oxidative DNA damage, and a recent study has shown that a positive feedback loop between ROS production and DNA damage response establishes senescence with the contribution of p21 (Passos et al, 2010). Although ROS are reported to mediate p53-dependent cell cycle arrest (Kulawiec et al, 2009), the mechanism by

[1]Biosignal Research Center, Kobe University, Kobe, Japan   [2]Department of Biology, Graduate School of Science, Kobe University, Kobe, Japan   [3]Department of Biology, Faculty of Science, Kobe University, Kobe, Japan   [4]Department of Bioresource Science, Graduate School of Agricultural Science, Kobe University, Kobe, Japan

Correspondence: skamada@kobe-u.ac.jp

which p53 regulates ROS production in the process of senescence induction remains mostly unclear. We have recently identified *DAO* to be up-regulated specifically in senescent cells and shown the direct transcriptional regulation of *DAO* by p53 (Nagano et al, 2016). Although we have revealed that ectopic expression of DAO inhibited proliferation of normal and tumor cells, it is still unknown whether and how DAO functionally contributes to the senescence process.

In the present study, we evaluated the functional association of DAO with the senescence process. We revealed that DAO accelerates senescence via enzymatic generation of ROS and that D-arginine, a substrate for DAO, is abundantly present in cultured cancer cells. DAO is activated in response to DNA damage presumably due to an increase in availability of its coenzyme, FAD.

# Results

## DAO promotes DNA damage– and oncogene-induced senescence through its enzymatic activity

To investigate whether DAO contributes to the senescence program, we evaluated the effect of *DAO* knockdown on DNA damage–induced senescence using two human tumor cell lines expressing the wild-type p53, osteosarcoma U2OS cells, and hepatocarcinoma HepG2 cells (Fig 1). U2OS and HepG2 cells transfected with two different siRNAs for *DAO* (DAO-1 and DAO-2) were treated with etoposide, an anticancer drug that induces DNA double-strand breaks (Wozniak & Ross, 1983), and the efficacy of *DAO* knockdown was confirmed by quantitative PCR (qPCR) and immunoblot analysis (Fig 1A and B). Because we could not reproducibly detect the DAO protein by immunoblot analysis in U2OS cells possibly because of its low expression level, we validated *DAO* knockdown by qPCR. In both cell lines, the DAO expression level was up-regulated in response to a sublethal dose of etoposide (2 $\mu$M and 10 $\mu$M, respectively) as observed previously (Nagano et al, 2016), which was effectively abolished by the treatment with siRNAs. Next, the extent of etoposide-induced senescence in the *DAO*-depleted cells was measured by two widely used senescence markers, senescence-associated $\beta$-galactosidase (SA-$\beta$-gal) staining (Dimri et al, 1995) and loss of proliferative capacity determined by colony-formation assay (Fig 1C–E). We observed that the percentage of SA-$\beta$-gal–positive cells was increased after the treatment with etoposide both in U2OS and HepG2 cells (Fig 1C and D), which is consistent with previous observations by us and others (Nakano et al, 2013; Nagano et al, 2016; te Poele et al, 2002). More importantly, siRNA-mediated depletion of *DAO* impaired the etoposide-induced SA-$\beta$-gal activation in both cell lines. Furthermore, knockdown of DAO partially restored etoposide-induced loss of proliferative capacity (Fig 1E). Moreover, immunoblot analysis showed that etoposide-induced up-regulation of p21, a critical mediator of senescence, was also impaired by *DAO* knockdown, although siRNAs of DAO-1 and DAO-2 may cause weak cellular stress because the p21 levels were slightly increased even in the absence of etoposide (Fig 1F), raising the possibility that DAO plays a role in inducing senescence. To confirm this, we next tested the impact of inhibiting DAO activity on senescence using 6-chlorobenzo[d]isoxazol-3-ol (CBIO), previously

characterized for the ability to inhibit enzymatic activity of DAO (Ferraris et al, 2008). CBIO suppressed the SA-$\beta$-gal activation and loss of proliferative capacity following etoposide treatment in both U2OS and HepG2 cells (Fig 2A–D). In accordance, immunoblot analysis revealed that etoposide-induced up-regulation of the p53 phosphorylation level at Ser15, which is crucial for p53 stabilization, and of the p53 and p21 protein levels were impaired by the CBIO treatment in U2OS and HepG2 cells, confirming the DAO role in senescence (Fig 2E and F). To further explore these findings, we examined the CBIO effect on etoposide-induced senescence in normal human fibroblast Hs68 cells. CBIO was also effective in suppressing senescence of Hs68 cells, as judged by SA-$\beta$-gal, EdU incorporation proliferation assays, and immunoblot analysis (Fig 2G–I), which suggests that the DAO role in senescence is common in both tumor and normal cells. In addition, oncogenic Ras–induced senescence of normal human fibroblast WI-38 cells was also impaired by the treatment with CBIO (Fig 2J–L), all implying the general role of DAO in the regulation of senescence.

To further assess the DAO role in senescence, we introduced an expression vector carrying *DAO* into U2OS cells (Fig 3A). We observed that ectopic expression of DAO enhanced etoposide-induced senescence, although DAO overexpression in the absence of etoposide had no significant effect (Fig 3B and C), indicating that DAO expression alone is not sufficient to induce senescence but accelerates senescence under stress conditions. In addition, unlike wild-type (wt) DAO, a DAO mutant with a point mutation (R199W), which was shown to have little enzymatic activity (<10% activity) in vivo and in vitro (Mitchell et al, 2010), was unable to promote senescence both in the absence and presence of etoposide (Fig 3). These results suggest that DAO facilitates senescence via its enzymatic activity.

## D-arginine is abundantly present in HepG2 cells

DAO exhibits broad substrate specificity, accepting neutral and polar D-amino acids as substrates (D'Aniello et al, 1993; Molla et al, 2006). To examine which of the DAO substrates can mediate the senescence-promoting effect of DAO, we measured the intracellular levels of seven major DAO substrates (D-alanine, D-arginine, D-aspartic acid, D-phenylalanine, D-proline, D-serine, and D-tyrosine) in HepG2 cells. Interestingly, considerably high levels of free D-arginine were observed in both etoposide-treated and untreated HepG2 cells, which were close to the levels of L-arginine, whereas the other D-amino acids were hardly detected (Table 1). The concentration of L-alanine, which can be used as an internal standard because of its constant abundance, was increased by 1.3-fold in cells treated with etoposide, and all amino acids examined were increased at similar rates to L-alanine (1.2- to 1.5-fold), indicating that the composition of these amino acids was largely unchanged during senescence. These data suggest that at least in this case, D-arginine can function as a substrate for DAO.

## D-arginine and D-serine enhances the senescence-promoting effect of DAO

We next set out to determine which of the substrates could function in the DAO-mediated senescence promotion. First, the

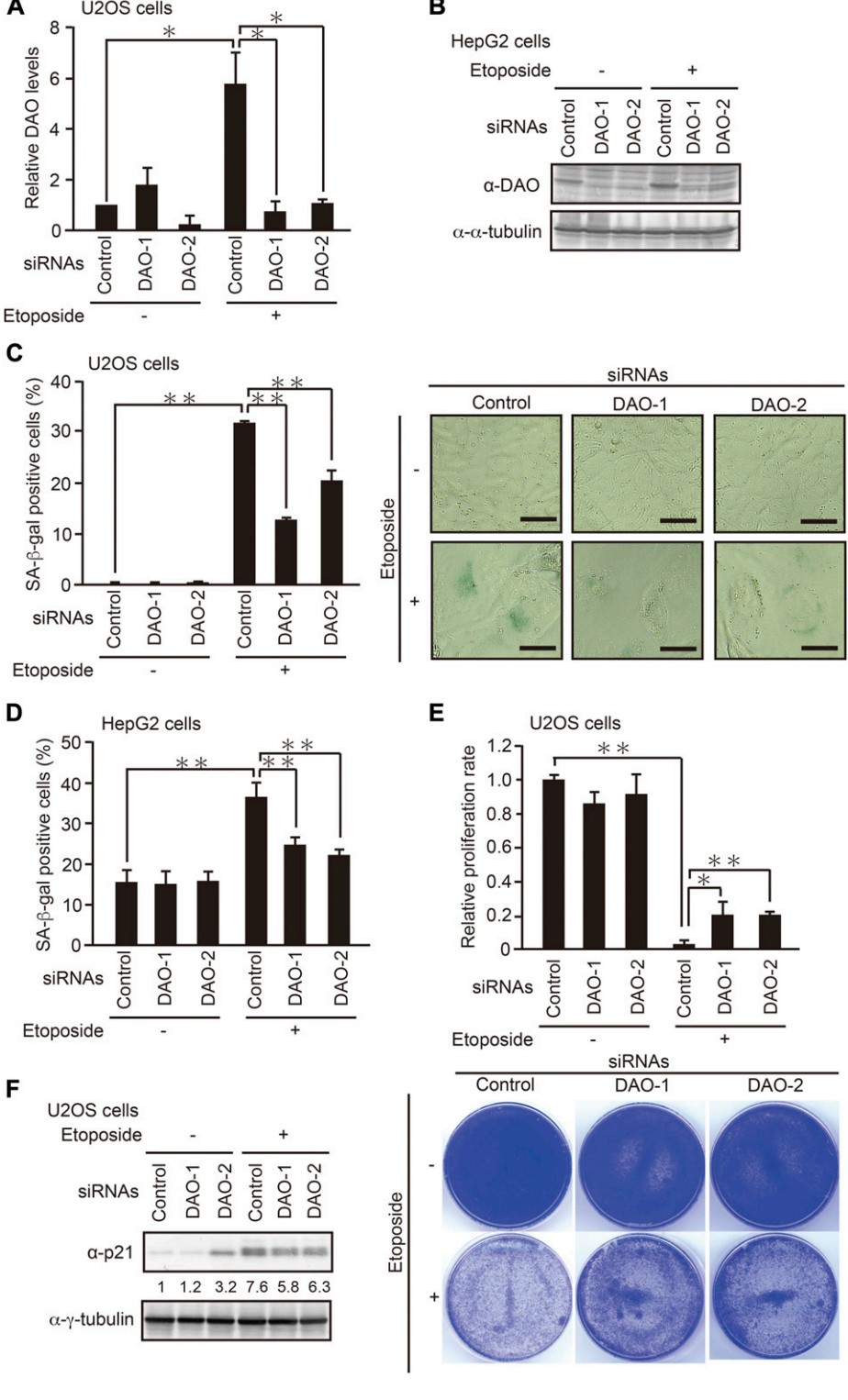

**Figure 1. Knockdown of *DAO* inhibits DNA damage–induced senescence.**
**(A)** U2OS cells transfected with siRNAs for *DAO* (DAO-1 and DAO-2) were treated with 2 μM etoposide for 7 d, and the expression levels of *DAO* were determined by qPCR. **(B)** HepG2 cells transfected with siRNAs for *DAO* (DAO-1 and DAO-2) were treated with 10 μM etoposide for 48 h, and the expression levels of DAO were determined by immunoblot analysis. **(C, D)** U2OS (C) and HepG2 (D) cells depleted of *DAO* were treated with 2 and 10 μM etoposide for 7 d and 48 h, respectively, and subjected to SA-β-gal staining. The percentage of SA-β-gal–positive cells (C left panel, D) and representative microscopic images (C right panel) are shown. Bars, 50 μm. **(E)** U2OS cells depleted of *DAO* were treated with 2 μM etoposide for 7 d and subjected to colony-formation assay. Relative proliferation rate (upper panel) and representative images (lower panel) are shown. **(F)** U2OS cells treated as in (E) but for 2 d instead of 7 d were subjected to immunoblot analysis. The protein levels relative to the γ-tubulin levels were quantified using NIH ImageJ software and are indicated at the bottom of each lane. Data are mean ± SD (*n* = 3 except in (A) where *n* = 2 independent experiments). Statistical significance is shown using the *t* test analysis; *P < 0.05, **P < 0.01.

DAO-overexpressing cells were treated with each of seven possible DAO substrates (ᴅ-alanine, ᴅ-arginine, ᴅ-methionine, ᴅ-phenylalanine, ᴅ-proline, ᴅ-serine, and ᴅ-tyrosine) in the presence of etoposide, and the extent of senescence was measured by performing SA-β-gal (Fig 4A) and EdU incorporation proliferation assays (Fig 4B).

Among these ᴅ-amino acids, ᴅ-arginine and ᴅ-serine potentiated the DAO effect on senescence promotion both with respect to SA-β-gal (Fig 4A) and EdU assays (Fig 4B). We also observed that the addition of ᴅ-arginine and ᴅ-serine by itself did not affect senescence induction (Fig 4C and D; compare bars 5 and 9 with bar 1).

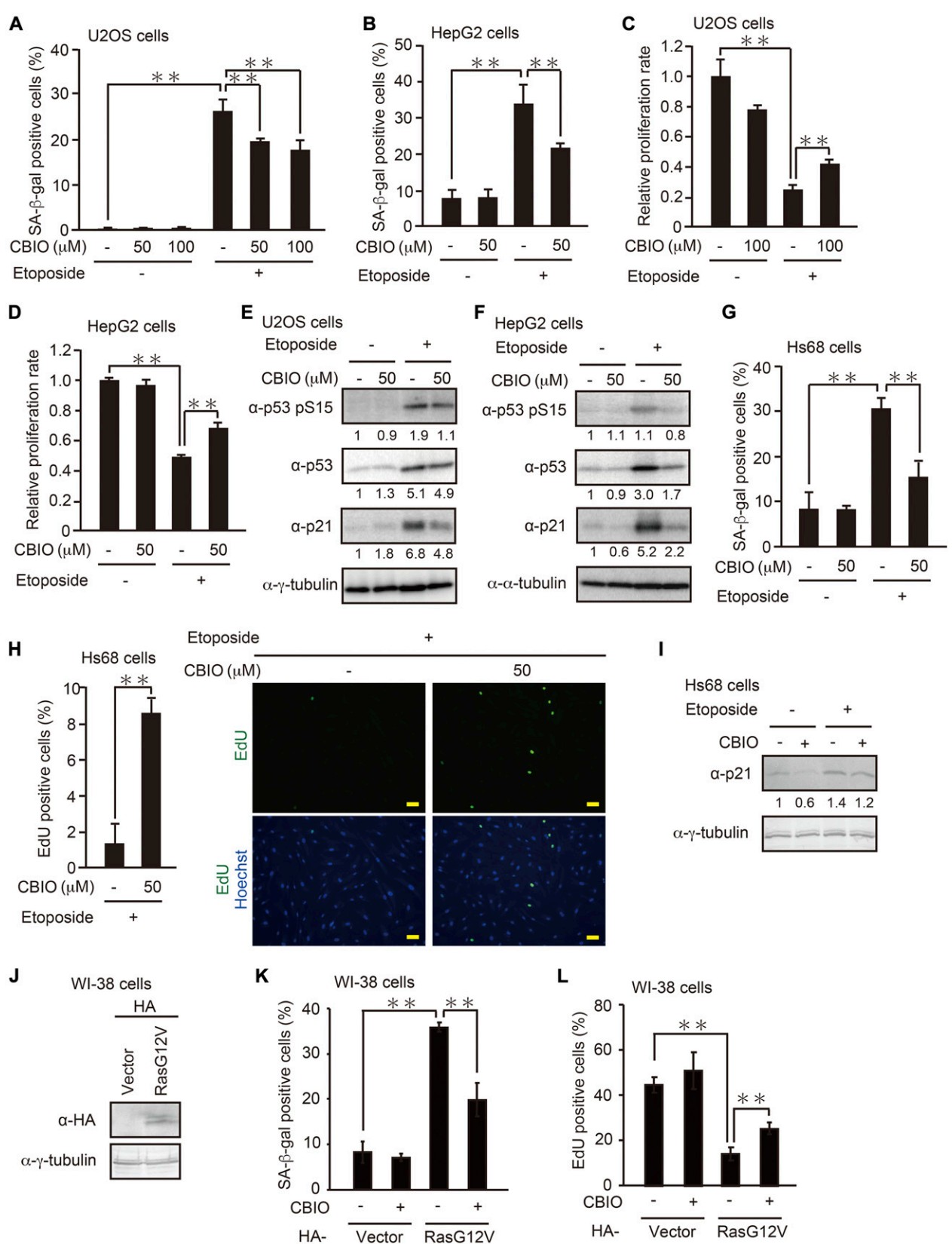

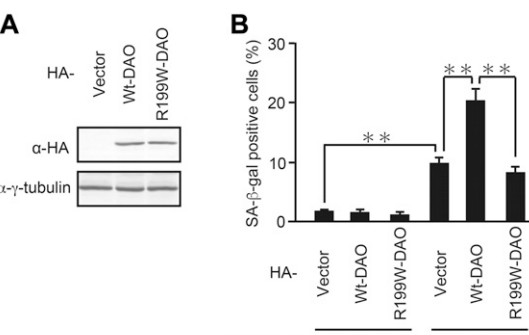

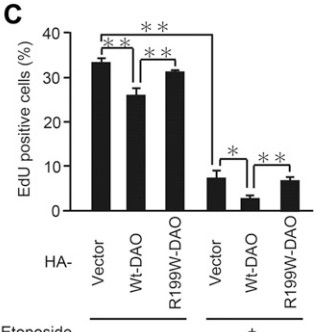

**Figure 3. Ectopic expression of wt-DAO, but not the inactive mutant, promotes senescence.**
**(A)** U2OS cells were transfected with pcDNA3-HA containing wt- and R199W-*DAO*, and the protein expression of DAO was confirmed by immunoblot analysis. **(B, C)** U2OS cells transfected as in (A) were selected with 800 µg/ml G418 and treated with 2 µM etoposide. After incubation for 7 d, the cells were subjected to SA-β-gal staining (B) and EdU incorporation assay (C). Data are mean ± SD (*n* = 3 independent experiments). Statistical significance is shown using the *t* test analysis; *P < 0.05, **P < 0.01.

Furthermore, the treatment with ʟ-arginine and ʟ-serine did not enhance the senescence promotion by DAO (Fig 4C and D; compare bars 13 and 14 with bar 4), indicating the specific effect of ᴅ-forms of arginine and serine. These results suggest that ᴅ-arginine and ᴅ-serine can mediate the senescence-promoting effect of DAO. We further noted here that the dual treatment of the ᴅ-amino acids and etoposide fully induced senescence to the level of the triple treatment with ᴅ-amino acids, etoposide, and DAO over-expression (Fig 4C and D; compare bar 6 with bar 8, and bar 10 with bar 12), whereas the combined treatment with ᴅ-amino acids and DAO overexpression hardly induced senescence (Fig 4C and D; bars 7 and 11). These results suggest that the high DAO expression and the substrate ᴅ-amino acids are not enough to induce se-nescence and that there is another regulatory factor that activates DAO in response to DNA damage.

### Riboflavin treatment activates DAO in the absence of DNA damage

What is the mechanism underlying the activation of DAO upon DNA damage? DAO is a flavoenzyme that uses FAD as a coenzyme (Rivlin, 1970; Depeint et al, 2006; Pollegioni et al, 2007), and we have revealed that *RFVT1* (also known as *GPR172B* or *SLC52A1*), a plasma membrane transporter of riboflavin, is up-regulated 45-fold in response to DNA damage in HepG2 cells (Nagano et al, 2016). Be-cause riboflavin is the essential precursor of FAD, we hypothesized that upon DNA damage, an increase in intracellular FAD level re-sults in the activation of DAO and thereby renders DAO capable of inducing senescence. Actually, we found that the intracellular FAD level was increased in response to etoposide treatment, which was abolished by knockdown of *RFVT1* (Fig 5A and B). To further test the hypothesis that the elevation of FAD level leads to DAO activation, we examined the effect of riboflavin treatment on senescence

of the DAO-overexpressing cells. As shown in Fig 5C and D, the combination of DAO overexpression and the riboflavin treatment markedly induced senescence despite the absence of etoposide treatment (compare bars 1 with 4). Furthermore, the addition of ᴅ-serine facilitated this senescence-inducing effect to a similar extent as in the etoposide-treated cells (i.e., there was no signif-icant additive effect of etoposide on senescence induced by the triple treatment with DAO overexpression, riboflavin, and ᴅ-serine; compare bars 6 with 8 of Fig 5C and D), suggesting that DAO can fully exert its senescence-inducing effect in the presence of both FAD and the appropriate substrate. In line with this, the combined treatment with riboflavin and ᴅ-serine induced up-regulation of *IL-6*, a key senescence-associated secretory phenotype factor, and DNA damage in DAO-overexpressing cells, as determined by qPCR and the immunostaining assay with 53BP1, a widely used DNA damage marker, respectively (Fig 5E and F). These results suggest that the DAO activity in the normal (i.e., pre-DNA damage) state is limited by the lack of FAD.

### DAO accelerates senescence through ROS production

We next set out to determine the molecular mechanism by which DAO promotes senescence. Because it is known that DAO produces ROS as byproducts of ᴅ-amino acid oxidation (Pollegioni et al, 2007) and that ROS induce senescence through oxidative DNA damage (Lu & Finkel, 2008; Kuilman et al, 2010; Campisi, 2013; Salama et al, 2014), we hypothesized that DAO promotes senescence via ROS generation. To test this, we monitored ROS levels in the *DAO*-depleted cells by using a fluorescent ROS indicator dye, CM-H$_2$DCFDA (Bass et al, 1983; Touyz & Schiffrin, 2001). As reported by others, we observed that ROS levels were increased in response to etoposide (Fig 6A). More importantly, knockdown of *DAO* partially

**Figure 2. Pharmacological inhibition of DAO impairs DNA damage– and oncogene-induced senescence.**
**(A–D)** U2OS (A, C) and HepG2 (B, D) cells treated with etoposide in the presence of 50 or 100 µM CBIO were subjected to SA-β-gal staining (A, B) and colony-formation assay (C, D). **(E, F)** U2OS (E) and HepG2 (F) cells were treated with 2 and 10 µM etoposide, respectively, for 2 d in the presence of 50 µM CBIO, and the expression levels of the indicated proteins were determined by immunoblot analysis. The protein levels relative to the γ-tubulin levels, except for the phosphorylated p53 (p53 pS15), which was normalized to the total p53 levels, were quantified using NIH ImageJ software and are indicated at the bottom of each lane. **(G, H)** Hs68 cells treated with 0.5 µM etoposide for 7 d in the presence of 50 µM CBIO were subjected to SA-β-gal staining (G) and EdU incorporation assay (H). The percentage of EdU-positive cells (H, left panel) and representative microscopic images (H right panel) are shown. Bars, 50 µm. **(I)** Hs68 cells treated with 0.5 µM etoposide for 2 d in the presence of 50 µM CBIO were subjected to immunoblot analysis. The protein levels were quantified as in (E, F). **(J)** WI-38 cells were transfected with pcDNA3-HA containing oncogenic *Ras*G12V, and the protein expression of RasG12V was confirmed by immunoblot analysis. **(K, L)** WI-38 cells transfected as in (J) were selected with 300 µg/ml G418 and treated with 50 µM CBIO. After incubation for 8 d, the cells were subjected to SA-β-gal staining (K) and EdU incorporation assay (L). Data are mean ± SD (*n* = 3 independent experiments). Statistical significance is shown using the *t* test analysis; **P < 0.01.

**Table 1. Concentrations of free D-/L-amino acids in HepG2 cells treated with 10 μM etoposide for 48 h.**

| | Form | Concentration (μM) | | Ratio (Etoposide/DMSO) |
|---|---|---|---|---|
| | | DMSO | Etoposide | |
| Ala | D | N/D | N/D | — |
| | L | 7791 | 10152 | 1.30 |
| Arg | D | 100 | 130 | 1.3 |
| | L | 120 | 160 | 1.33 |
| Asp | D | Trace | Trace | — |
| | L | 624 | 1006 | 1.61 |
| Phe | D | N/D | N/D | — |
| | L | 328 | 437 | 1.33 |
| Pro | D | N/D | N/D | — |
| | L | 2033 | 2439 | 1.20 |
| Ser | D | Trace | Trace | — |
| | L | 2312 | 3068 | 1.33 |
| Tyr | D | N/D | N/D | — |
| | L | 380 | 570 | 1.5 |

N/D, not detected; trace, detected but below limit of quantification.

suppressed the etoposide-induced ROS accumulation with a statistical significance, although the effects were modest (Fig 6A). Similar results were obtained when DAO was inhibited by CBIO in U2OS and HepG2 cells (Fig 6B and C). On the other hand, overexpression of DAO strongly potentiated the etoposide-induced increase in ROS levels, yet DAO overexpression alone resulted in only a slight elevation of ROS (Fig 6D), which is consistent with the observation that DAO overexpression alone was insufficient to induce senescence (Fig 3B and C). Furthermore, treatment with N-acetyl-L-cysteine (NAC), a potent ROS scavenger, suppressed senescence induced by DAO overexpression to a similar extent in the cells transfected with empty vector, suggesting that the promotion of senescence by DAO is completely dependent on enzymatic ROS formation (Fig 6E–G). Consistent with this, combined treatment with NAC and CBIO had no synergistic effect on senescence repression when compared with the single treatment with NAC (Fig 6H–J). These results suggest that DAO promotes senescence fully in an ROS-dependent manner and DAO is partly responsible for ROS accumulation during senescence.

Because PRODH, another flavoenzyme, has been recently reported to activate p53-mediated senescence through enzymatic ROS production (Nagano et al, 2017), we tested whether pharmacological inhibition of DAO and PRODH exerts an additive effect on senescence suppression. The combination of CBIO and L-tetrahydro-2-furoic acid (THFA), a specific inhibitor for PRODH (Tallarita et al, 2012), had a more marked effect on inhibiting senescence than the single treatment with either inhibitor alone in both tumor U2OS and normal Hs68 cells (Fig 6K–P). These results suggest that DAO and PRODH cooperatively contribute to the ROS generation responsible for senescence induction.

These results collectively demonstrate that DAO up-regulation in response to DNA damage results in ROS accumulation, thereby enhancing oxidative stress and the subsequent p53-p21 signaling pathway, and ultimately promotes senescence.

# Discussion

Although the amounts of free D-amino acids in mammalian tissues are extremely low in most cases, it is becoming clear that D-amino acids and their metabolic enzymes have distinct biological functions. The best-known example is D-serine–mediated activation of NMDA receptors (Schell et al, 1995; Katsuki et al, 2004; Billard, 2012). Besides the brain, DAO is highly expressed in the kidney and liver in mammals, but much less attention has been devoted to investigating the DAO roles in other tissues and systems. In particular, the relationship between DAO and senescence is yet to be described. We have recently reported that the *DAO* gene is transcriptionally activated by p53 in response to senescence-inducing stimuli, but our previous study did not fully explore the direct relationship between DAO and senescence (Nagano et al, 2016). In the present study, we found that genetic and pharmacological inhibition of DAO abrogated DNA damage–induced senescence, whereas ectopic expression of DAO promoted it. Furthermore, DAO overexpression increased intracellular ROS levels, and the senescence-promoting effect of DAO was abolished by treatment with NAC, an ROS scavenger, indicating that acceleration of senescence by DAO is dependent on ROS production.

ROS are recognized as critical initiators of senescence, and their accumulation is widely observed in senescent cells (Lu & Finkel, 2008; Kuilman et al, 2010; Campisi, 2013; Salama et al, 2014). It has been reported that several ROS-producing enzymes, such as PRODH and NADPH oxidases, act to promote senescence (Geiszt et al, 2000; Nagano et al, 2017; Jun & Lau, 2010). In line with these findings, we observed that inhibition of DAO alone partially (i.e., not fully) suppressed etoposide-induced ROS accumulation and senescence and that combined treatment with CBIO and THFA, a potent inhibitor of PRODH, had an additive effect on senescence suppression. These results suggest that ROS produced by DAO alone are insufficient to initiate the senescence program and that DAO works cooperatively with other ROS-producing enzyme(s) to increase the intracellular ROS level beyond a threshold triggering senescence. In addition, because the etoposide-induced up-regulation of p53 and p21 protein levels was impaired by the CBIO treatment in U2OS and HepG2 cells (Fig 2E and F), we would like to claim the formation of a positive feedback loop, in which DNA damage activates p53, which in turn transactivates DAO-producing ROS, finally leading to further activation of p53 through oxidative stress and enhancement of senescence. Moreover, we should emphasize here that only DAO overexpression did not induce senescence, but when combined with etoposide, it promoted etoposide-induced senescence, implying the DNA damage–induced mechanism of DAO activation. Given that the D-amino acid levels are unchanged during senescence (Table 1), it is likely that DAO activity in itself is potentiated by DNA damage.

In striking contrast to DAO, it has been shown that overexpression of PRODH, another flavin-dependent oxidase, by itself induces senescence through enzymatic ROS production (Nagano et al, 2017). What causes this difference between DAO and PRODH? One possible explanation is the difference in subcellular localization. Whereas DAO is localized in peroxisome (Pollegioni et al, 2007), PRODH is localized in mitochondria (Phang et al, 2008). It has

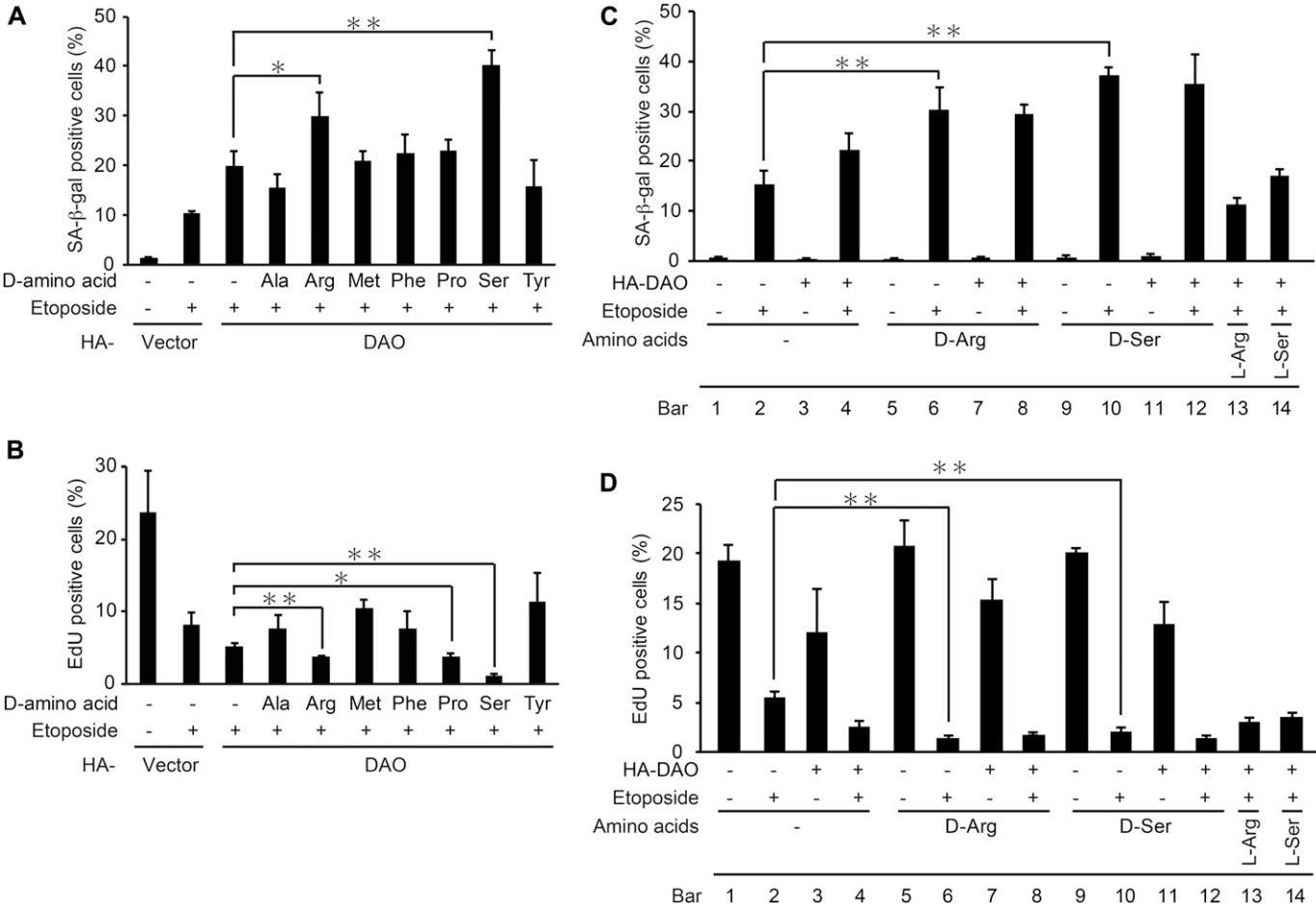

**Figure 4.** ᴅ-arginine and ᴅ-serine enhances the senescence-promoting effect of DAO.
**(A, B)** U2OS cells were transfected with pcDNA3-HA containing wt-*DAO*, selected with 800 μg/ml G418, and treated with each of seven ᴅ-amino acids at 5 mM (except for ᴅ-tyrosine which was used at 2.5 mM) in the presence of 2 μM etoposide as indicated. After incubation for 7 d, the cells were subjected to SA-β-gal staining (A) and EdU incorporation assay (B). **(C, D)** U2OS cells were transfected as in (A, B) in the presence of ᴅ-arginine, ᴅ-serine, ʟ-arginine, or ʟ-serine at 5 mM for 7 d and subjected to SA-β-gal staining (C) and EdU incorporation assay (D). Data are mean ± SD ($n$ = 3 independent experiments). Statistical significance is shown using the $t$ test analysis; *$P < 0.05$, **$P < 0.01$.

been revealed by a subcellular fractionation study that FAD is abundantly present in mitochondria and cytoplasm but less in peroxisome (Leighton et al, 1982), which raises the possibility that the DAO activity in the absence of etoposide is limited because of the low FAD concentration in peroxisome. FAD is a redox cofactor of a large number of oxidases, dehydrogenases, and reductases (termed "flavoenzymes") that are compartmentalized in appropriate cellular organelles (Rivlin, 1970; Depeint et al, 2006), and DAO is a peroxisome-localized one (Pollegioni et al, 2007). In cells, FAD is enzymatically produced from its precursor, riboflavin, through FAD synthase that catalyzes the FAD biosynthesis in both cytoplasm and mitochondria, but proteins responsible for FAD synthesis in peroxisome have not been described to date (Torchetti et al, 2010). This compartment-specific biosynthesis seems to reflect the intracellular demand of FAD. For example, FAD is essential for β-oxidation of fatty acids in mitochondria (Depeint et al, 2006), whereas peroxisomal fatty acid β-oxidation requires oxygen, but not FAD, as an electron acceptor to form hydrogen peroxide. Considering this intracellular demand and supply mechanism, the

FAD concentration in peroxisome appears to be relatively low compared with other subcellular compartments. In fact, Leighton et al (1982) have shown that the peroxisomal FAD level is about 8 and 13 times less than that in mitochondria and cytoplasm, respectively, in rat liver (Leighton et al, 1982), which leads us to speculate that the DAO activity is regulated by peroxisomal FAD level. Consistently, we found that when combined with riboflavin treatment, overexpression of DAO induced senescence. These results suggest that DAO could not fully exert its enzymatic activity owing to the low level of peroxisomal FAD in the absence of DNA damage, but upon DNA damage, peroxisomal FAD level is elevated to the extent that DAO can become activated and promote senescence. In support of this model, we have previously found a plasma membrane riboflavin transporter, RFVT1 (also known as GPR172B or SLC52A1), to be up-regulated in response to DNA damage (Nagano et al, 2016). Furthermore, in the present study, we observed that the intracellular FAD level was increased upon DNA damage in an RFVT1-dependent manner. However, it is still an open question how DAO activity is regulated upon DNA damage, because we have

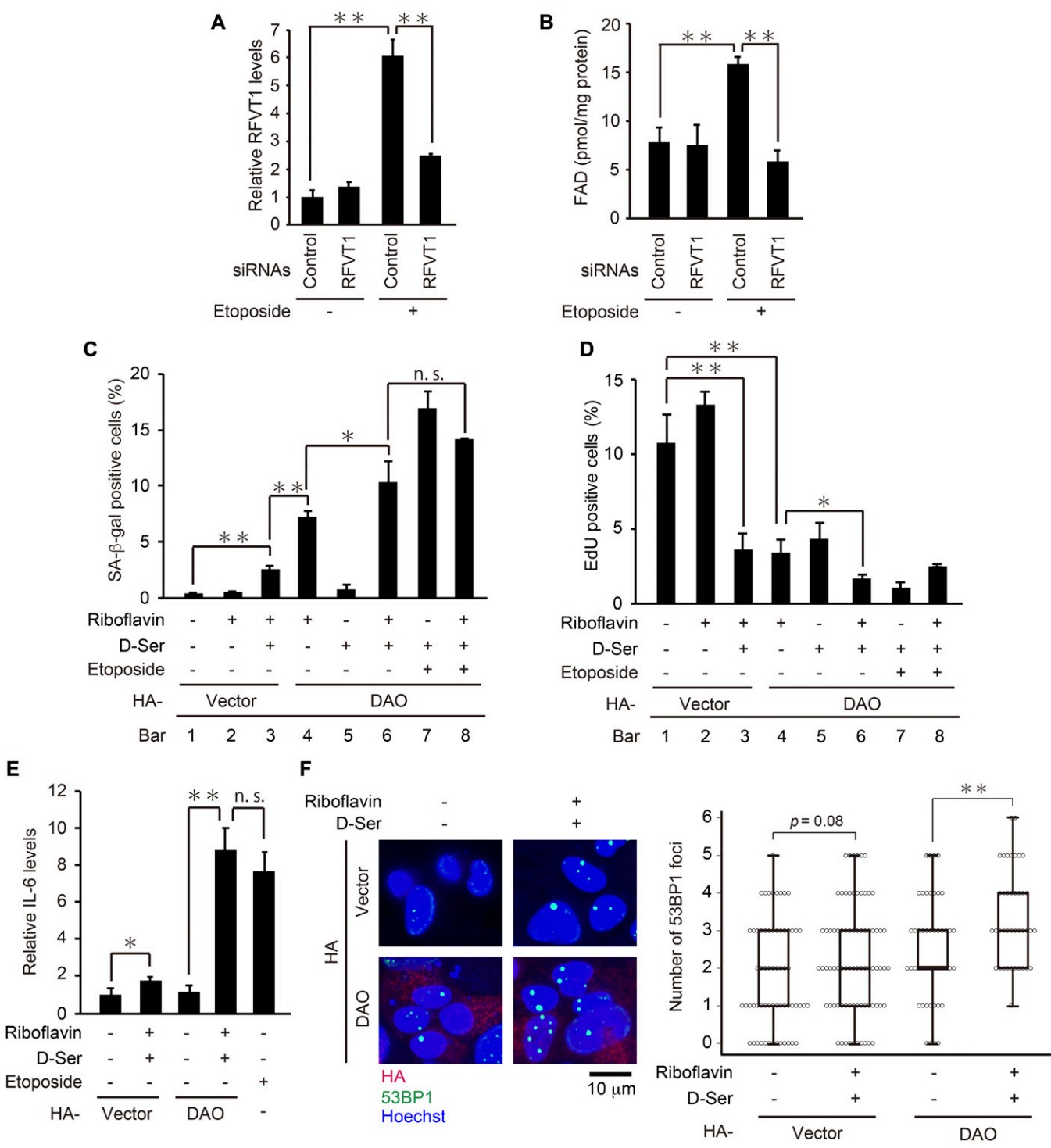

**Figure 5. Riboflavin treatment activates the DAO activity in the absence of DNA damage.**
**(A)** U2OS cells transfected with siRNA for *RFVT1* were treated with 2 μM etoposide for 7 d, and the expression levels of *RFVT1* were determined by qPCR. **(B)** U2OS cells depleted of *RFVT1* were treated with 2 μM etoposide for 7 d and subjected to FAD quantification. The concentrations of FAD per mg protein of the cells are shown. **(C, D)** U2OS cells were transfected with pcDNA3-HA containing wt-*DAO*, selected with 800 μg/ml G418, and treated with 50 μM riboflavin and 5 mM D-serine in the presence of 2 μM etoposide as indicated. After incubation for 7 d, the cells were subjected to SA-β-gal staining (C) and EdU incorporation assay (D). **(E)** U2OS cells were treated as in (C, D), and the expression levels of *IL-6* were determined by qPCR. **(F)** U2OS cells overexpressing DAO were treated with 50 μM riboflavin and 5 mM D-serine for 7 d and subjected to immunostaining for 53BP1, HA, and Hoechst staining. Representative microscopic images (left) and box plots of the number of 53BP1 foci in HA-DAO–expressing cells (right) are shown. The upper and lower limits of the boxes and the lines across the boxes indicate the 75th and 25th percentiles and the median, respectively. Error bars (whiskers) indicate the 90th and 10th percentiles, respectively. Statistical significance (*P*-value) is shown using the *t* test analysis (*n* = 50 cells). Data are mean ± SD (*n* = 3 independent experiments). Statistical significance is shown using the *t* test analysis; *P < 0.05, **P < 0.01, and n.s.: not significant (*P* > 0.05).

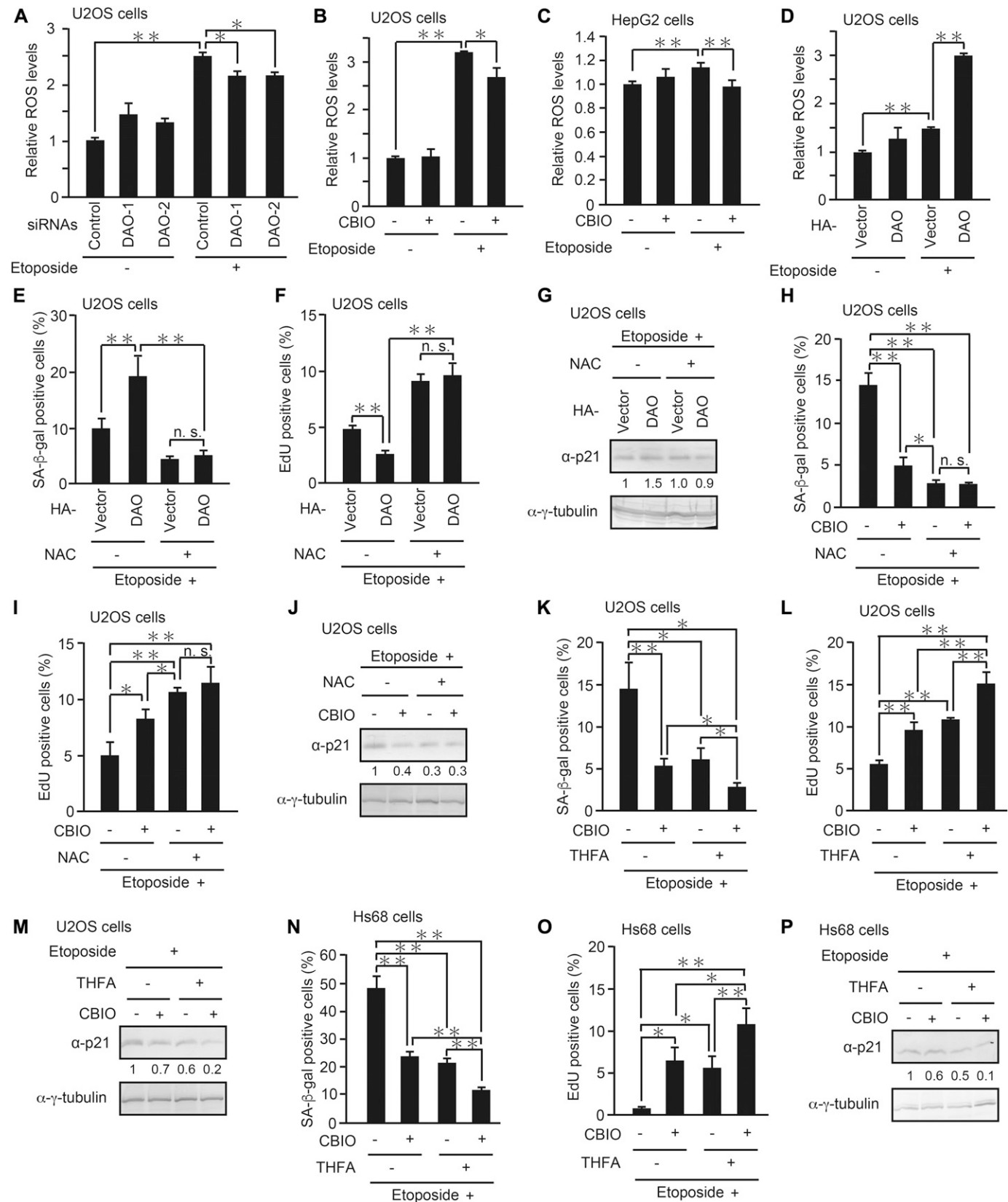

not compared the peroxisomal FAD levels between normal and DNA-damaged cells yet. Further studies are needed to determine whether the DAO activity is controlled by peroxisomal FAD level or not.

Consistent with the DAO role in ROS production, antitumor therapies with the introduction of DAO have been reported (Fang et al, 2002, 2004; El Sayed et al, 2012a, b). El Sayed et al have demonstrated that stable expression of DAO enhances the anti-cancer effect of 3-bromopyruvate, a hexokinase II inhibitor that interferes with glycolysis, through induction of oxidative stress. Because senescence can be a potent tumor suppressive mechanism, the novel relationship between DAO and senescence provides new insights into the DAO role as a tumor suppressor, that is, DAO can inhibit tumor formation not merely through the promotion of oxidative stress but through the induction of senescence, which may hopefully facilitate the development of new therapeutic strategies and applications for cancer.

As mentioned previously, D-amino acids are considered to be much less abundant than L-amino acids in mammals. Surprisingly, however, we found that D-arginine was abundantly present in HepG2 cells, whose level was comparable with that of L-arginine. D-arginine has been reported to be present in the brain and to act as a protector of the central nervous system against neurotoxicity induced by high levels of glucocorticoids (Canteros, 2014). However, to our knowledge, no studies have described the presence of D-arginine in cultured cells, and hence, its biological role is presently uncertain. In the present study, we showed that the addition of D-arginine and D-serine facilitated the DAO effect on senescence promotion, suggesting their possible role as the substrates of DAO in DNA damage–induced senescence. Although further analysis is needed to determine whether D-arginine and D-serine mediate the effect of DAO on senescence in vitro and in vivo, our results may shed new light on the role of these D-amino acids. At the same time, we do not exclude the possibility that DAO metabolizes D-amino acids other than those tested in our present study because DAO can oxidize a wide range of D-amino acids including D-norleucine and D-ornithine (D'Aniello et al, 1993; Molla et al, 2006). Actually, so far, we have been unable to detect D-arginine and D-serine at quantifiable levels in normally proliferating U2OS cells, suggesting that the D-amino acid(s) required for DAO-mediated senescence promotion varies depending on the cell type and/or on the senescence stage. The answers to these issues will provide a better understanding of the roles of D-amino acids in biological processes.

Senescence is now considered to play a critical role in age-related pathologies and in tumor suppression, resulting from the permanent loss of cell proliferation capacity (Kuilman et al, 2010;

Campisi, 2013; Salama et al, 2014; Yanai & Fraifeld, 2017). The DAO protein is up-regulated not only in premature but also in replicative senescence (Nagano et al, 2016), and several studies have reported that the amounts of D-amino acids increase with age through spontaneous racemization of L-amino acids, yet their physiological effects on senescence and aging are still ambiguous (Thorpe et al, 2010; Chervyakov et al, 2011). Therefore, whether or not DAO is involved in these biological processes in vivo is the key question for future research.

In summary, our results clearly show a novel function of DAO as a promoter of DNA damage–induced senescence, which may provide new insights into the roles of D-amino acids in various physiological and pathological processes including senescence, cancer, and aging.

# Materials and Methods

## Cell culture, treatment, and transfection

U2OS (a human osteosarcoma line; ATCC), Hs68 (normal human diploid fibroblasts; IFO50350, JCRB Cell Bank) (Sayles & Johnson, 1996), and WI-38 (normal human diploid fibroblasts; RCB0702, provided by the RIKEN BRC through the National Bio-Resource Project of the MEXT/AMED, Japan) cells were cultured in DMEM (Wako) containing 10% FBS. HepG2 (a human hepatocellular carcinoma line; a gift from Dr. S Shimizu) cells were cultured in RPMI 1640 medium (Wako) containing 10% FBS. For induction of senescence, U2OS and Hs68 cells were treated with etoposide (Sigma-Aldrich) at 2 and 0.5 $\mu$M, respectively, for 48 h and cultured in the medium without the drug for additional 5 d to develop senescent phenotypes and used for subsequent analyses whereas HepG2 cells were treated with 10 $\mu$M etoposide for 48 h. The concentrations and durations of etoposide necessary to induce senescence were determined previously (Nakano et al, 2013; Nagano et al, 2016, 2017). To inhibit DAO, the cells were incubated with 50 or 100 $\mu$M CBIO (EMD Millipore). Amino acids were obtained from the following sources: D-alanine, D-arginine, D-phenylalanine, D-proline, and D-tyrosine from Sigma-Aldrich; D-methionine and D-serine from Wako; L-arginine and L-serine from Nacalai Tesque. Riboflavin (Nacalai Tesque) was used at 50 $\mu$M. To scavenge ROS, the cells were incubated with 1 mM NAC (Sigma-Aldrich). When THFA (Sigma-Aldrich) was used, the cells were incubated with the reagent at 5 mM. Transfection with expression vectors was carried out using FuGENE HD (Promega) according to the manufacturer's instructions. Where

**Figure 6.    DAO enhances DNA damage–induced ROS accumulation, cooperating with PRODH.**
**(A)** U2OS cells transfected with siRNAs for *DAO* (DAO-1 and DAO-2) were treated with 2 $\mu$M etoposide for 2 d, and the ROS level was measured. **(B, C)** U2OS (B) and HepG2 (C) cells treated with 2 and 10 $\mu$M etoposide, respectively, for 2 d in the presence of 50 $\mu$M CBIO were subjected to ROS assay. **(D)** U2OS cells transfected with pcDNA3-HA-DAO and treated with 2 $\mu$M etoposide were subjected to ROS assay. **(E, F)** U2OS cells overexpressing DAO were treated with etoposide in the presence of 1 mM NAC for 7 d and subjected to SA-$\beta$-gal staining (E) and EdU incorporation assay (F). **(G)** U2OS cells treated as in **(E, F)** but for 2 d instead of 7 d were subjected to immunoblot analysis. The protein levels relative to the $\gamma$-tubulin levels were quantified using NIH ImageJ software and are indicated at the bottom of each lane. **(H, I)** U2OS cells treated with etoposide in combination with 50 $\mu$M CBIO and 1 mM NAC as indicated for 7 d were subjected to SA-$\beta$-gal staining (H) and EdU incorporation assay (I). **(J)** U2OS cells treated as in (H, I) but for 2 d instead of 7 d were subjected to immunoblot analysis. The protein levels were quantified as in (G). **(K, L)** U2OS cells treated as in (H, I) but 5 mM THFA instead of NAC were subjected to SA-$\beta$-gal staining (K) and EdU incorporation assay (L). **(M)** U2OS cells treated as in (K, L) but for 2 d instead of 7 d were subjected to immunoblot analysis. The protein levels were quantified as in (G). **(N, O)** Hs68 cells treated as in (K, L) were subjected to SA-$\beta$-gal staining (N) and EdU incorporation assay (O). **(P)** Hs68 cells treated as in (N, O) but for 2 d instead of 7 d were subjected to immunoblot analysis. The protein levels were quantified as in (G). Data are mean ± SD ($n$ = 3 independent experiments). Statistical significance is shown using the $t$ test analysis; *$P$ < 0.05, **$P$ < 0.01, and n.s. = not significant ($P$ > 0.05).

indicated, the transfectants were selected with G418 (Wako) at 800 μg/ml (U2OS cells) or 300 μg/ml (WI-38 cells) for 5 d. Stock solutions of etoposide and CBIO were prepared in DMSO, whereas riboflavin, NAC, THFA, and the amino acids were dissolved in water.

### RNA interference

ON-TARGETplus human *DAO* siRNA (DAO-1, J-009756-08 and DAO-2, J-009756-09), SMARTpool ON-TARGETplus *RFVT1* siRNA (L-010712-0005), and their control siRNA (D-001810-10) were from GE Healthcare Dharmacon. U2OS and HepG2 cells were seeded and transfected with 30 nM siRNA using Lipofectamine RNAiMAX Transfection Reagent (Thermo Fisher Scientific) according to the manufacturer's instructions.

### RNA isolation and qPCR

Total RNA was isolated from siRNA-transfected U2OS cells using RNeasy Mini Kit (QIAGEN), and cDNA was synthesized using ReverTra Ace qPCR RT Master Mix with gDNA Remover (TOYOBO) according to the manufacturer's instructions. The resulting cDNAs were subjected to qPCR (LightCycler480 Real-Time PCR System; Roche Applied Science) using specific primers for *DAO* (a forward primer: 5′-CCACTGGA-CATAAAGGTCTACG-3′ and a reverse primer: 5′-GGGTTGTTGGGGTCA-GAAAG-3′), *GAPDH* (a forward primer: 5′-CAATGACCCCTTCATTGACCT-3′ and a reverse primer: 5′-ATGACAAGCTTCCCGTTCTC-3′), *RFVT1* (a forward primer: 5′-TTGCTGTTGCCATCACTACC-3′ and a reverse primer: 5′-CAAAGCCTCTTCTTCCTCCTTC-3′), and *IL-6* (a forward primer: 5′-GGGCACCTCAGATTGTTGTT-3′ and a reverse primer: 5′-GTGCCCAGTG-GACAGGTTT-3′). Relative expression levels were calculated by ΔΔCt methods (ΔCt sample-ΔCt calibrator).

### Antibodies

Anti-p53 antibody (DO-1; sc-126), HRP-conjugated anti-rat antibody (sc-2032), and AP-conjugated anti-rat antibody (sc-2021) were obtained from Santa Cruz Biotechnology; HRP-conjugated anti-rabbit antibody (W4011), HRP-conjugated anti-mouse antibody (W4021), and AP-conjugated anti-mouse antibody (S3721) were from Promega; anti-α-tubulin antibody (T9026) and anti-γ-tubulin antibody (T6557) were from Sigma-Aldrich; anti-phospho-p53 (Ser15) antibody (#9284) and anti-53BP1 antibody (#4937) were from Cell Signaling Technology; anti-DAO antibody (6844-1) was from Abcam; anti-HA antibody (1867423) was from Roche; anti-p21 antibody (K0081-3) was from Medical and Biological Laboratories; and AP-conjugated anti-rabbit antibody (59298) was from MP Biomedicals.

### Immunoblot analysis

The cells were lysed in SDS sample buffer (50 mM Tris–HCl [pH 6.8], 2% SDS, 5% 2-mercaptoethanol, 0.1% bromophenol blue, and 10% glycerol), and the lysates were separated by SDS–polyacrylamide gel electrophoresis and blotted onto Immobilon polyvinylidene difluoride membrane (EMD Millipore). Each protein was visualized using primary antibodies, corresponding enzyme-conjugated secondary antibodies (HRP or AP), and the chemiluminescent ECL (GE Healthcare) or the chromogenic NBT/BCIP (Nacalai Tesque) substrates.

### SA-β-gal assay

SA-β-gal assay was performed using Senescence β-Galactosidase Staining Kit (Cell Signaling Technology) according to the manufacturer's instructions. In brief, the cells were fixed with 2% formaldehyde/0.2% glutaraldehyde and incubated with SA-β-Gal staining solution (1 mg/ml 5-bromo-4-chloro-3-indolyl-β-D-galac-toside, 40 mM citric acid/sodium phosphate [pH 6.0], 5 mM potassium ferrocyanide, 5 mM potassium ferricyanide, 150 mM NaCl, and 2 mM $MgCl_2$) for 24 h. The stained cells were observed under fluorescence microscope (model BZ-8000; Keyence). Senescent cells were identified as blue-stained cells, and at least 100 cells in randomly selected microscopic fields were counted to determine the percentage of SA-β-gal–positive cells.

### Colony-formation assay

For colony-formation assay, $2 × 10^4$ cells (U2OS cells) or $4 × 10^4$ cells (HepG2 cells) were plated in a 35-mm dish, cultured for 3-7 d, and stained with crystal violet (Wako). The purple stained area was measured using NIH ImageJ software and expressed as relative proliferation rate.

### EdU incorporation assay

The cells were labeled with EdU for 24 h (Hs68 and WI-38 cells) or 3 h (U2OS cells) before fixation, and then EdU incorporation was detected using Click-iT EdU Imaging Kit (Life Technologies) according to the manufacturer's instructions. After staining the nuclei with 10 μM Hoechst 33342, the cells were observed under fluorescence microscope (model BZ-9000; KEYENCE).

### Plasmid constructions

To construct pcDNA3-HA-RasG12V, an expression vector that contain full-length human G12V-H-Ras, the human *Ras*G12V cDNA was amplified with a pair of primers (a forward primer: 5′-GC-GAATTCATGACGGAATATAAGCTGGTG-3′ and a reverse primer: 5′-GCAGCGGCCGCTCAGGAGAGCACACACTTG-3′) using a vector, pCMV-HA-RasG12V (kindly provided by Dr. K Kaibuchi, Nagoya University, Japan), as a template. The resulting fragment was digested with EcoRI and NotI and cloned into downstream of the HA tag sequence in the pcDNA3-HA vector (Invitrogen). For construction of pcDNA3-HA-DAO, an expression vector of HA-tagged human full-length DAO (NCBI accession number: NM_001917.4), the *DAO* cDNA was amplified with a pair of primers (a forward primer: 5′-CGTGCTCGGAATTCATGCGTGTGGTGGTGATTGGAG-3′ and a reverse primer: 5′-CGTGCTCGGCGGCCGCTCAGAGGTGGGATGGTGGCATT-3′) using a cDNA sample prepared from U2OS cells as a template, and the resulting fragments were digested with Eco RI and Not I, and cloned into downstream of the HA tag sequence in the pcDNA3-HA vector. To generate a catalytically inactive DAO mutant (R199W), PCR reactions were performed using mutagenic primers (a forward primer: 5′-CCCCTGCTGCAGCCAGGCTGGGGGCAGATCATGAAGG-3′ and a reverse primer: 5′-CCTTCATGATCTGCCCCCAGCCTGGCTGCAGCAGGGG-3′).

## Measurement of intracellular free amino acid concentrations

Concentrations of amino acids were determined by a two-dimensional HPLC system combining a microbore-monolithic ODS column and a narrowbore-enantioselective column (Miyoshi et al, 2009; Hamase et al, 2010) performed by Shiseido Corporation. Briefly, the cells were homogenized in 90% methanol and centrifuged to obtain supernatants. After derivatization of amino acids with a fluorescence derivatizing reagent, NBD-F, the samples were injected into the HPLC.

## FAD assay

FAD assay was performed using FAD Assay Kit (ab204710; Abcam) according to the manufacturer's instructions. Briefly, the cells were lysed and deproteinized with perchloric acid, and the resulting samples were mixed with the reaction mixture. The concentration of FAD was measured by a colorimetric assay based on the FAD-dependent reaction of OxiRed probe using a spectrophotometer (iMark microplate reader; Bio-Rad).

## Immunostaining

For immunostaining analysis, the cells were fixed with 4% paraformaldehyde and permeabilized in 0.5% Triton X-100, then incubated with anti-53BP1 and anti-HA antibodies overnight at 4°C followed by incubation with the Alexa Fluor 488- and 546-conjugated secondary antibodies (Life Technologies) for 1 h at room temperature. After staining the cell nuclei with Hoechst 33342, the cells were examined under fluorescence microscope (model BZ-9000; KEYENCE).

## ROS assay

ROS levels were determined by using CM-$H_2$DCFDA (Thermo Fisher Scientific) following the manufacturer's protocol. Briefly, the cells were incubated in PBS containing 1 $\mu$M CM-$H_2$DCFDA for 10 min at 37°C. Afterwards, the cells were trypsinized and suspended in PBS, and the fluorescence was measured using a BD Accuri C6 flow cytometer (BD Biosciences).

## Statistical analysis

$t$ test was used to calculate $P$-values for all datasets.

# Acknowledgements

We thank Dr. K Kaibuchi (Nagoya University, Japan) for providing the pCMV-HA-RasG12V plasmid. We also thank Mr. Yuto Awai and Ms. Mizuki Kinugasa for their cordial technical support. This work was supported by Japan Society for the Promotion of Science (JSPS) KAKENHI Grant Numbers 25640063 and 17K15595.

## Author Contributions

T Nagano: conceptualization, formal analysis, funding acquisition, investigation, and writing—original draft, review, and editing.
S Yamao: conceptualization, formal analysis, and investigation.
A Terachi: formal analysis and investigation.
H Yarimizu: formal analysis and investigation.
H Itoh: resources, formal analysis, and investigation.
R Katasho: formal analysis and investigation.
K Kawai: formal analysis and investigation.
A Nakashima: formal analysis and resources.
T Iwasaki: formal analysis and resources.
U Kikkawa: supervision and resources.
S Kamada: conceptualization, formal analysis, supervision, funding acquisition, project administration, and writing-original draft, review, and editing.

## Conflict of Interest Statement

The authors declare that they have no conflict of interest.

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
