## [Reviewer comments · Life Science Alliance]

Life Science Alliance

D-amino acid oxidase promotes cellular senescence via the production of reactive oxygen species

Taiki Nagano, Shunsuke Yamao, Anju Terachi, Hidetora Yarimizu, Haruki Itoh, Ryoko Katasho, Kosuke Kawai, Akio Nakashima, Tetsushi Iwasaki, Ushio Kikkawa, and Shinji Kamada
DOI: 10.26508/lsa.201800045

Corresponding author(s): Shinji Kamada, Kobe University

Review Timeline:	Submission Date:	2018-03-06
	Editorial Decision:	2018-04-03
	Revision Received:	2018-09-26
	Editorial Decision:	2018-10-10
	Revision Received:	2019-01-08
	Editorial Decision:	2019-01-11
	Revision Received:	2019-01-11
	Accepted:	2019-01-11

Scientific Editor: Andrea Leibfried

Transaction Report:

April 3, 2018

Re: Life Science Alliance manuscript #LSA-2018-00045-T

Prof. Shinji Kamada
Kobe University
Biosignal Research Center
1-1 Rokkodai-cho, Nada-ku
Kobe, Hyogo 657-8501
Japan

Dear Dr. Kamada,

Thank you for submitting your manuscript entitled "D-amino acid oxidase promotes cellular senescence via the production of reactive oxygen species" to Life Science Alliance. Your manuscript has now been reviewed by three referees whose comments are included below.

As you will see, the referees overall appreciate your work. However, numerous questions were also raised, importantly, regarding the generalisation of your conclusions and the assays used to monitor senescence, as well as regarding the support provided for the proposed pathway underlying DAO-dependent senescence. We therefore think that further experiments are required to make your manuscript a strong candidate for publication. If you think that the manuscript can be modified according to the constructive input provided by the referees, we would be happy to consider a revised manuscript for publication in Life Science Alliance.

In considering a revised manuscript, we suggest focusing on the following items:

1. To compliment the existing data demonstrating senescence using SA- β -gal and proliferation assays, two referees suggest adding an additional marker, for example expression analysis of a cell cycle inhibitor (e.g. p16).
2. Two referees suggest examining the requirement of DAO for additional forms of senescence induction, for example oncogene-induced senescence.
3. Examine if cellular or peroxisomal FAD levels increase during senescence induction, and/or if RFVT is required for senescence.
4. Referee 1 calls into question the significance of the data shown in Figure 5C-E, as the effects of the cytosolic-localized mutant appear to be modest. We agree with the referee, and you may consider removing these experiments from the manuscript.

Should you be able to address the issues raised, you can upload the revised version of your manuscript by logging in to your account: <https://lsa.msubmit.net/cgi-bin/main.plex>
You will be guided to complete the submission of your revised manuscript and to fill in all necessary information.

While you are revising your manuscript, please also attend to the following editorial points to help

expedite the publication of your manuscript. Please direct any editorial questions to the journal office.

-- High-resolution figure, supplementary figure and video files uploaded as individual files: See our detailed guidelines for preparing your production-ready images, <http://life-science-alliance.org/authorguide>

B. MANUSCRIPT ORGANIZATION AND FORMATTING:

Full guidelines are available on our Instructions for Authors page, <http://life-science-alliance.org/authorguide>

Thank you for this interesting contribution to Life Science Alliance. We are looking forward to receiving your revised manuscript.

Sincerely,

Reviewer #1 (Comments to the Authors (Required)):

Nagano et al. have investigated the role of D-amino acid oxidase (DAO, which oxidizes neutral and polar D-amino acids) in DNA damage-induced cellular senescence (DDIS). This group previously reported that DAO is upregulated in senescent cells in a p53-dependent manner, although its function was not addressed. Here they show that knockdown of DAO or inhibition of its activity reduces senescence and partially prevents growth arrest. DAO overexpression enhances senescence in a manner that requires its enzymatic activity, but only in the context of DNA damage. The authors demonstrate that HepG2 cells contain appreciable levels of D-Arginine, a DAO substrate. Accordingly, addition of D-Arg or D-Ser to the culture media increased DDIS. Nagano et al. suggest that DAO acts by increasing intracellular ROS, a product of the oxidation reaction involving specific D-amino acids, and propose that elevated DAO activity in senescent cells may be dependent on increased availability of its co-factor, FAD.

The role of DAO in senescence is a noteworthy finding and the experiments showing its involvement are convincing. Nevertheless, supporting details such as a full analysis of senescence features regulated by DAO and a demonstration that DAO activity increases in senescent cells are lacking and require further experimental evidence. Overall, the mechanistic basis for the pro-senescence functions of DAO in the DDIS model is not fully developed. These and other points are listed below.

Specific comments:

1. In this study senescence is assessed solely by analyzing SA-bGal activity and cell proliferation. However, additional markers such as p16 induction and expression of a panel of senescence-associated secretory phenotype (SASP) genes (e.g., pro-inflammatory cytokines and growth factors) also should be measured in control and DAO depleted cells, with or without etoposide treatment. These experiments would help to illuminate which features of senescence are under DAO control.
2. It would be of interest to investigate whether oncogene-induced senescence also requires DAO. This would address the generality of the DAO pathway in senescence induction.
3. The authors' results suggest that DAO expression alone does not stimulate senescence but rather requires a DNA damage signal. On this basis, they postulate that increased DAO activity, possibly involving elevated peroxisomal FAD, promotes senescence. To prove this model, they should measure the specific enzymatic activity of DAO in lysates from normal and senescent cells.

4. It is also possible that DAO activity is not stimulated by DDR signaling except through its increased expression. The dependence on etoposide seen for the senescence-stimulating effect of ectopic DAO (Figures 3B and C) could easily be explained by a requirement for p53 and its other effectors, which are clearly also critical for senescence induction.

5. Figure 2E and F: it is stated that p53 and p21 protein levels are "remarkably impaired" by treatment with the DAO inhibitor, CBIO. However, the immunoblots show rather modest decreases in levels of these proteins, particularly in U2OS cells. Also, given that DAO is believed to be a p53 target gene and is therefore downstream of p53, why should the DAO inhibitor affect p53 activation? Do the authors believe that p53 activation is ROS-dependent and therefore is not directly induced by etoposide-mediated DNA damage? If so, perhaps p53 activation would be suppressed by NAC. Finally, since the above conclusions are based on use of the inhibitor, CBIO, which could have off-target effects, the authors should replicate these experiments using siRNA to deplete DAO.

6. The analysis of the cytoplasmic DAO-deltaC1 mutant (Figure 5) is not particularly definitive or informative. The pro-senescence activity of this mutant is quite similar to that of WT DAO (Figures 5C and D). A more revealing experiment would be to measure peroxisomal FAD levels to determine if they increase in senescence cells. If so, this would suggest a basis for increased DAO enzymatic activity, which presumably contributes to senescence induction in DDIS cells.

7. The effects of DAO knockdown on ROS levels (Figure 6) are quite modest. Therefore, the reviewer is not convinced that increased ROS mediates the pro-senescence activity of DAO. Also, ROS levels in HepG2 cells are barely altered by etoposide treatment (Figure 6C). It is possible that another species such as lipid ROS is critical for senescence induction and this is also neutralized by NAC, explaining the results in Figures 6G and H.

8. Minor point: several figures include both raw image data as well as quantitative graphs of the results. The images could be presented the first time the assay is used and omitted thereafter, saving space and creating room for additional data in the figures.

Reviewer #2 (Comments to the Authors (Required)):

In this study, the authors suggest that D-amino acid oxidase (DAO) is up-regulated in DNA-damage-induced cells whose activity promotes senescence induction through elevating the level of reactive oxygen species (ROS). They demonstrate that although the high level of DAO per se is insufficient for the induction of senescence, the increased level of riboflavin may be also required for the induction. Finally, they showed that the expression of PRODH, another flavoenzyme, is induced in senescent cells, which collaborates with DAO to promote senescence.

Most of the experiments in this manuscript were well-done, and the results largely supported their conclusions. However, several important issues have to be addressed before publication. Especially, the regulatory mechanisms underlying the production of high level FAD in senescent cells and their generality are missing. If these issues can be adequately addressed, the paper will be suitable for publication in the journal.

Major comments:

1. Authors suggested that FAD level is up-regulated in senescent cells, possibly due to the induction of RFVT, and this up-regulation is required for DAO-induced senescence. They should

demonstrate the experimental evidence that FAD is actually up-regulated in senescent cells and RFVT has a crucial role in DAO-induced senescence.

2. Authors performed all experiments using etoposide-induced senescence. Therefore, it is very difficult to draw clear conclusions that the authors' observations are general mechanisms of senescent induction. Some important findings should be repeated using senescent cells induced by other stimuli, such as oncogene activation and replicative senescence.

Minor comments:

1. Authors indicated that DAO-wt localized to peroxisomes, but the mutants localized to cytosol. Co-staining with peroxisome marker is required to clarify this point.

Reviewer #3 (Comments to the Authors (Required)):

This manuscript by Nagano et al. has revealed that D-amino acid oxidase (DAO)-mediated ROS production promotes the induction of cellular senescence, and the activity of DAO is regulated by the availability of its substrate (D-arginine, D-serine) and co-enzyme (FAD). This work is not only an extension of the previous study published in Scientific Report by the same group but reveals the underlying mechanisms linking the DAO-mediated D-amino acid metabolism and cellular senescence. Therefore, this manuscript may provide us an important point of view about the glucose metabolism and the amino acid metabolism, which may play a crucial role in the induction of cellular senescence. In this regard, this manuscript is potentially interesting. However, significantly more work is needed to make this paper suitable for publication.

(1) In this manuscript, the authors have used the percentage of SA- β -gal positive cell and EdU incorporation inhibited cells as judge of senescence induction. However, it has been reported that the knockdown of lysosomal β -galactosidase (GLB), which is an essential protein of SA- β -gal, did not interfere with senescence (Lee et al., Aging Cell, 2006). Therefore, SA- β -gal activity seems not the necessary factor for senescence. On the other hand, senescence is defined as the irreversible cell cycle arrest that can be induced by CDK inhibitors, p16 or p21 mediated DNA damage signaling. Consequently, the data of DNA damage foci and expression of CDK inhibitors are needed for detection of cellular senescence in all figures.

(2) I'm wondering why the expression level of DAO was significantly decreased when cells were treated by Etoposide, although siRNA of DAO-1 was not efficient under normal condition without Etoposide in U2OS cells in Figure 1A.

(3) In Figure 1E, the proliferation data was only shown in U2OS cell. The same effect of DAO knockdown has been detected in HepG2 cells?

(4) The authors mention that p53 phosphorylation level at Ser15 was impaired by the CBIO treatment in U2OS (Fig 2E). However, it seems no difference between control and CBIO treated cells. The authors should explain why.

(5) In Table 1, the authors have shown that the D-arginine is rich in HepG2 cells. Then, how about the concentration of amino acids in other cell types?

(6) In Fig4A, SA- β -gal positive cell was increased when only treated with Riboflavin and D-serine (compare bars 1 and 3, 4 and 6). Could DNA damage be also detected in these cells?

In considering a revised manuscript, we suggest focusing on the following items:

1. To compliment the existing data demonstrating senescence using SA- β -gal and proliferation assays, two referees suggest adding an additional marker, for example expression analysis of a cell cycle inhibitor (e.g. p16).

According to the reviewers' constructive comments, we have compared expression levels of p21, a cell cycle inhibitor, under various conditions. The results show that the p21 expression largely correlated with the extent of senescence determined by SA- β -gal and proliferation assays, which complements our findings that DAO has the causal role in promoting senescence. These new data have been added as Figures 1F, 2I, and 6G, J, M, and P, and the manuscript has been modified accordingly.

2. Two referees suggest examining the requirement of DAO for additional forms of senescence induction, for example oncogene-induced senescence.

Based on the suggestions of these reviewers, we have tested the DAO role in oncogenic Ras-induced senescence of WI-38 cells, a normal human fibroblast cell line (Hs68 cells were not used for this analysis owing to a low transfection efficiency). The results show that overexpression of RasG12V induced senescence in WI-38 cells, which was impaired by the treatment with a DAO inhibitor, CBIO. These results suggest that DAO has a general role in the regulation of senescence regardless of stimuli type. These new data have been added as Figure 2J-L, and the manuscript has been modified accordingly.

3. Examine if cellular or peroxisomal FAD levels increase during senescence induction, and/or if RFVT is required for senescence.

According to the instruction, we have examined the cellular FAD levels and found that the intracellular FAD level was elevated during senescence induction, which was abolished by knockdown of RFVT1. This result clearly indicates that the cellular FAD level increases in senescent cells in an RFVT1-dependent manner. These new data have been added as Figure 5A and B, and the manuscript has been modified accordingly.

4. Referee 1 calls into question the significance of the data shown in Figure 5C-E, as

the effects of the cytosolic-localized mutant appear to be modest. We agree with the referee, and you may consider removing these experiments from the manuscript.

We also agree that these data have only a modest significance. Therefore, we have removed these data, and the manuscript has been modified accordingly.

Reviewer #1 (Comments to the Authors (Required)):

Nagano et al. have investigated the role of D-amino acid oxidase (DAO, which oxidizes neutral and polar D-amino acids) in DNA damage-induced cellular senescence (DDIS). This group previously reported that DAO is upregulated in senescent cells in a p53-dependent manner, although its function was not addressed. Here they show that knockdown of DAO or inhibition of its activity reduces senescence and partially prevents growth arrest. DAO overexpression enhances senescence in a manner that requires its enzymatic activity, but only in the context of DNA damage. The authors demonstrate that HepG2 cells contain appreciable levels of D-Arginine, a DAO substrate. Accordingly, addition of D-Arg or D-Ser to the culture media increased DDIS. Nagano et al. suggest that DAO acts by increasing intracellular ROS, a product of the oxidation reaction involving specific D-amino acids, and propose that elevated DAO activity in senescent cells may be dependent on increased availability of its co-factor, FAD.

The role of DAO in senescence is a noteworthy finding and the experiments showing its involvement are convincing. Nevertheless, supporting details such as a full analysis of senescence features regulated by DAO and a demonstration that DAO activity increases in senescent cells are lacking and require further experimental evidence. Overall, the mechanistic basis for the pro-senescence functions of DAO in the DDIS model is not fully developed. These and other points are listed below.

Specific comments:

1. In this study senescence is assessed solely by analyzing SA-bGal activity and cell proliferation. However, additional markers such as p16 induction and expression of a panel of senescence-associated secretory phenotype (SASP) genes (e.g.,

pro-inflammatory cytokines and growth factors) also should be measured in control and DAO depleted cells, with or without etoposide treatment. These experiments would help to illuminate which features of senescence are under DAO control.

We wish to thank the reviewer for the comment. Inspired by the recommendations of this reviewer and the editor as well as by the critiques raised by the Reviewer #3 (Major point 1), we have analyzed an additional senescence marker, p21 expression, under various conditions including DAO knockdown, since p16 is known not to be expressed in U2OS cells. The results show that the p21 expression mostly correlated with the extent of senescence determined by SA- β -gal and proliferation assays, which supports our conclusion that DAO has the causal role in promoting senescence. These new data have been added as Figure 1F, 2I, and 6G, J, M, and P.

2.It would be of interest to investigate whether oncogene-induced senescence also requires DAO. This would address the generality of the DAO pathway in senescence induction.

We thank the reviewer for this constructive comment. We have performed this experiment in oncogenic Ras-induced senescence of WI-38 cells, a normal human fibroblast cell line (Hs68 cells were not used for this analysis owing to a low transfection efficiency). The results show that overexpression of RasG12V induced senescence in WI-38 cells, which was impaired by the treatment with the DAO inhibitor, CBIO. These results suggest that DAO has a general role in the regulation of senescence regardless of types of senescence-inducing stimuli. These new data have been added as Figure 2J-L.

3.The authors' results suggest that DAO expression alone does not stimulate senescence but rather requires a DNA damage signal. On this basis, they postulate that increased DAO activity, possibly involving elevated peroxisomal FAD, promotes senescence. To prove this model, they should measure the specific enzymatic activity of DAO in lysates from normal and senescent cells.

We agree with the reviewer that it is important to measure the enzymatic activity of DAO. However, so far we have been unable to perform this experiment due to the unavailability of experimental system that allows quantitative monitoring of the DAO activity in our laboratory. Alternatively, we have measured FAD concentrations in

senescent cells and found that the FAD concentration is elevated during senescence. Given that DAO is a flavoenzyme utilizing FAD as a coenzyme, and that treatment with a FAD precursor, riboflavin, potentiated the senescence-promoting effect of DAO (Figure 5C and D), this result supports our idea that FAD elevation in response to DNA damage results in DAO activation. These new data have been added as Figure 5A and B. However, since it would remain important to measure DAO activity in normal and senescent cells, we plan to carry out this type of analysis in future.

4.It is also possible that DAO activity is not stimulated by DDR signaling except through its increased expression. The dependence on etoposide seen for the senescence-stimulating effect of ectopic DAO (Figures 3B and C) could easily be explained by a requirement for p53 and its other effectors, which are clearly also critical for senescence induction.

We appreciate the reviewer's comment on this point. We agree that DAO is not activated by DDR signaling itself but possibly stimulated by a mechanism that requires p53. The possible mechanism we proposed in this manuscript is a p53-dependent induction of riboflavin transporter, RFVT1. We have previously reported that RFVT1 is upregulated during senescence, which is dependent on p53 (Nagano et al, *Sci Rep* 6: 31758, 2016). As discussed above (Major point 3), DAO is a flavoenzyme using FAD as a coenzyme, and the intracellular FAD level was increased upon DNA damage in an RFVT1-dependent manner (Figure 5A and B). Furthermore, the treatment with a FAD precursor, riboflavin, potentiated the senescence-promoting effect of DAO (Figure 5C and D). These results support the mechanism that FAD elevation in response to DNA damage leads to DAO activation.

5.Figure 2E and F: it is stated that p53 and p21 protein levels are "remarkably impaired" by treatment with the DAO inhibitor, CBIO. However, the immunoblots show rather modest decreases in levels of these proteins, particularly in U2OS cells. Also, given that DAO is believed to be a p53 target gene and is therefore downstream of p53, why should the DAO inhibitor affect p53 activation? Do the authors believe that p53 activation is ROS-dependent and therefore is not directly induced by etoposide-mediated DNA damage? If so, perhaps p53 activation would be suppressed by NAC. Finally, since the above conclusions are based on use of the inhibitor, CBIO,

which could have off-target effects, the authors should replicate these experiments using siRNA to deplete DAO.

We wish to thank the reviewer for this comment. We agree that the sentence “p53 and p21 protein levels are remarkably impaired” is an overstatement, and thus we have removed the word “remarkably” from the original text (P. 7, lines 15-16). In addition, to clarify the difference in protein levels, the protein levels relative to the γ -tubulin levels have been quantified and indicated at the bottom of each lane (Figure 2E and F).

Regarding the relationships among the p53 activation, DNA damage, and ROS, our claim in this manuscript is that DNA damage activates p53, which in turn transactivates DAO. The transactivated DAO then produces ROS, finally leading to a further activation of p53 through oxidative stress and to an enhancement of senescence (*i.e.* forming a positive-feedback loop that ultimately promotes senescence). Consistent with this idea, the p53 protein level was indeed suppressed by the NAC treatment, as shown in Figure 1 in this rebuttal letter (please see below). To better explain our claim, we have added the following text in the Discussion section (P. 15, lines 7-12).

“In addition, since the etoposide-induced upregulation of p53 and p21 protein levels was impaired by the CBIO treatment in U2OS and HepG2 cells (Fig 2E and F), we would like to claim the formation of a positive-feedback loop, in which DNA damage activates p53, which in turn transactivates DAO producing ROS, finally leading to a further activation of p53 through oxidative stress and to an enhancement of senescence.”

Finally, with respect to the experiments using siRNA, we have analyzed the p21 protein levels in U2OS cells in which DAO was depleted by siRNA-mediated knockdown and found that the etoposide-induced p21 upregulation was impaired by DAO knockdown. This result is consistent with the result obtained using the DAO inhibitor, CBIO. This new data has been added as Figure 1F.

Figure 1. U2OS cells treated with etoposide in combination with 50 μ M CBIO and 1 mM NAC as indicated for 2 days were subjected to immunoblot analysis. The protein levels relative to the γ -tubulin levels were quantified using NIH ImageJ software and are indicated at the bottom of each lane. The p53 protein level was decreased by the treatment with NAC. This figure contains the data of Figure 6J of the manuscript (p21 and γ -tubulin blots).

6. The analysis of the cytoplasmic DAO-deltaC1 mutant (Figure 5) is not particularly definitive or informative. The pro-senescence activity of this mutant is quite similar to that of WT DAO (Figures 5C and D). A more revealing experiment would be to measure peroxisomal FAD levels to determine if they increase in senescence cells. If so, this would suggest a basis for increased DAO enzymatic activity, which presumably contributes to senescence induction in DDIS cells.

According to this comment as well as the suggestion raised by the editor, we have removed the section concerning the DAO- Δ C1 mutant. However, since it would be important to compare the peroxisomal FAD levels between normal and senescent cells, we plan to establish an appropriate experimental system to address this issue and intend to investigate in detail in future.

7. The effects of DAO knockdown on ROS levels (Figure 6) are quite modest. Therefore, the reviewer is not convinced that increased ROS mediates the pro-senescence activity of DAO. Also, ROS levels in HepG2 cells are barely altered by etoposide treatment (Figure 6C). It is possible that another species such as lipid ROS is critical for senescence induction and this is also neutralized by NAC, explaining the results in Figures 6G and H.

We agree with the reviewer that the effects of DAO knockdown on the ROS levels were modest. However, these effects were statistically significant as shown in Figure 6A. To clarify this point, we have added the following text in the Results section (P. 11, lines

19-20), “with a statistical significance, although the effects were modest”. This is the reason why we concluded that DAO is partially, but not solely, responsible for the ROS production during senescence (p12, lines 10-12). Our claim in this manuscript is that DAO partially contributes to senescence induction, and that DAO works cooperatively with other ROS-producing enzyme(s) (e.g. PRODH) to increase the intracellular ROS level beyond a threshold triggering senescence. But of course, we do not exclude the possibility that another mechanism (e.g. lipid ROS) also contributes to senescence induction. However, our results here show that DAO inhibition by siRNAs and by CBIO significantly impairs the ROS production and senescence induction, and therefore we wish to retain the original version of the manuscript.

8.Minor point: several figures include both raw image data as well as quantitative graphs of the results. The images could be presented the first time the assay is used and omitted thereafter, saving space and creating room for additional data in the figures.

We thank the reviewer for this comment. According to the instruction, we have omitted the raw images in Figures 1D, 2C and D, and 3C.

Reviewer #2 (Comments to the Authors (Required)):

In this study, the authors suggest that D-amino acid oxidase (DAO) is up-regulated in DNA-damage-induced cells whose activity promotes senescence induction through elevating the level of reactive oxygen species (ROS). They demonstrate that although the high level of DAO per se is insufficient for the induction of senescence, the increased level of riboflavin may be also required for the induction. Finally, they showed that the expression of PRODH, another flavoenzyme, is induced in senescent cells, which collaborates with DAO to promote senescence.

Most of the experiments in this manuscript were well-done, and the results largely supported their conclusions. However, several important issues have to be addressed before publication. Especially, the regulatory mechanisms underlying the production of high level FAD in senescent cells and their generality are missing. If these issues can be adequately addressed, the paper will be suitable for publication in the journal.

Major comments:

1. Authors suggested that FAD level is up-regulated in senescent cells, possibly due to the induction of RFVT, and this up-regulation is required for DAO-induced senescence. They should demonstrate the experimental evidence that FAD is actually up-regulated in senescent cells and RFVT has a crucial role in DAO-induced senescence.

We wish to thank the reviewer for this constructive critique. We have carried out these experiments, and found that the FAD level was upregulated in senescent cells, which was abolished by knockdown of RFVT1. These results demonstrate that FAD is actually upregulated in senescent cells and RFVT1 has a crucial role in the regulation of FAD level. These new data have added as Figure 5A and B.

2. Authors performed all experiments using etoposide-induced senescence. Therefore, it is very difficult to draw clear conclusions that the authors' observations are general mechanisms of senescent induction. Some important findings should be repeated using senescent cells induced by other stimuli, such as oncogene activation and replicative senescence.

We appreciated the reviewer's comment on this point. As suggested by the reviewer as well as by the reviewer #1 (Major point 2) and the editor, we have tested the DAO role in oncogenic Ras-induced senescence of WI-38 cells, a normal human fibroblast cell line (Hs68 cells were not used for this analysis owing to a low transfection efficiency). We observed that DAO inhibition by CBIO impaired the oncogene-induced senescence, which suggests the general role of DAO in the senescence induction. These new data have been added as Figure 2J-L.

Minor comments:

1. Authors indicated that DAO-wt localized to peroxisomes, but the mutants localized to cytosol. Co-staining with peroxisome marker is required to clarify this point.

We agree with the reviewer. However, according to the suggestions by the reviewer #1 (Major point 6) and the editor, we omitted the data concerning the cytosol-localized DAO mutant. Therefore, we will carry out the co-staining experiment in future.

Reviewer #3 (Comments to the Authors (Required)):

This manuscript by Nagano et al. has revealed that D-amino acid oxidase (DAO)-mediated ROS production promotes the induction of cellular senescence, and the activity of DAO is regulated by the availability of its substrate (D-arginine, D-serine) and co-enzyme (FAD). This work is not only an extension of the previous study published in Scientific Report by the same group but reveals the underlying mechanisms linking the DAO-mediated D-amino acid metabolism and cellular senescence. Therefore, this manuscript may provide us an important point of view about the glucose metabolism and the amino acid metabolism, which may play a crucial role in the induction of cellular senescence. In this regard, this manuscript is potentially interesting. However, significantly more work is needed to make this paper suitable for publication.

(1) In this manuscript, the authors have used the percentage of SA- β -gal positive cell and EdU incorporation inhibited cells as judge of senescence induction. However, it has been reported that the knockdown of lysosomal β -galactosidase (GLB), which is an essential protein of SA- β -gal, did not interfere with senescence (Lee et al., Aging Cell, 2006). Therefore, SA- β -gal activity seems not the necessary factor for senescence. On the other hand, senescence is defined as the irreversible cell cycle arrest that can be induced by CDK inhibitors, p16 or p21 mediated DNA damage signaling. Consequently, the data of DNA damage foci and expression of CDK inhibitors are needed for detection of cellular senescence in all figures.

We appreciate the reviewer's comment on this point. As discussed above (Reviewer #1, Major point 1), we have now shown that p21 expression largely correlated with the extent of senescence determined by SA- β -gal and EdU proliferation assays, supporting our conclusion that DAO plays the role in promoting senescence. These new data have been added as Figure 1F, 2I, and 6G, J, M, and P.

(2) I'm wondering why the expression level of DAO was significantly decreased when cells were treated by Etoposide, although siRNA of DAO-1 was not efficient under normal condition without Etoposide in U2OS cells in Figure 1A.

We agree that siRNA of DAO-1 did not decrease the mRNA expression of DAO in the normal (pre-DNA damage) state of U2OS cells. This is presumably because the basal expression level of DAO is kept at a considerably low level, and thus siRNA of DAO-1

was unable to further reduce the mRNA level in U2OS cells. Indeed, we could not reproducibly detect the DAO protein by immunoblot analysis in U2OS cells probably due to its low expression level (that is why we analyzed the DAO expression by qPCR in U2OS cells). Figure 1A shows that DAO expression was upregulated in response to etoposide, which was effectively inhibited by the treatment with the DAO-1 siRNA. Therefore, we concluded that the DAO-1 siRNA effectively inhibited the DAO expression under the condition where DAO was sufficiently expressed.

(3) In Figure 1E, the proliferation data was only shown in U2OS cell. The same effect of DAO knockdown has been detected in HepG2 cells?

The reviewer's comment is correct. Unfortunately, however, we have only the data obtained from the experiments in which DAO knockdown was achieved using an siRNA pool, a mixture of four different siRNAs (DAO-1, DAO-2, DAO-3, and DAO-4; DAO-1 and DAO-2 are the same siRNAs used in the main manuscript) in HepG2 cells. The results show that etoposide-induced loss of proliferative capacity of HepG2 cells was partially impaired by DAO knockdown as shown in Figure 2 in this rebuttal letter (please see below). This result is fundamentally consistent with the results in U2OS cells. However, due to the difference in experimental conditions, we do not intend to add this data to the manuscript and would like to retain the original version of the manuscript.

Figure 2. HepG2 cells transfected with an siRNA pool against DAO, comprising four different oligonucleotides including DAO-1 and DAO-2 (Dharmacon; L-009756-00-0005) were treated with 10 μ M etoposide for 48 h subjected to colony-formation assay. Relative proliferation rate is shown. Data are mean \pm s.d. ($n = 3$). Statistical significance is shown using the Student's t-test analysis; ** $p < 0.01$. Etoposide reduced the cell proliferation of HepG2 cells, which was impaired by DAO knockdown.

(4) The authors mention that p53 phosphorylation level at Ser15 was impaired by the CBIO treatment in U2OS (Fig 2E). However, it seems no difference between control and CBIO treated cells. The authors should explain why.

We agree that the difference of p53 phosphorylation levels between control and CBIO-treated cells is modest. To clarify this point, the p53 phosphorylation levels relative to the γ -tubulin level have been quantified using NIH ImageJ software and indicated at the bottom of each lane. As a result, the relative p53 phosphorylation level was elevated to 8.2 in response to etoposide treatment, which was decreased to 5.4 by the CBIO treatment (Figure 2E). Therefore, we concluded that the p53 phosphorylation level was impaired by the CBIO treatment to some extent, which is consistent with our claim that DAO partially contributes to (*i.e.* not completely responsible for) senescence induction.

(5) In Table 1, the authors have shown that the D-arginine is rich in HepG2 cells. Then, how about the concentration of amino acids in other cell types?

According to the reviewer's comment, we have measured the concentrations of 15 D-amino acids (alanine, arginine, asparagine, aspartic acid, cysteine, glutamine, glutamic acid, histidine, leucine, lysine, methionine, phenylalanine, serine, threonine, and valine) in normally proliferating U2OS cells, because we detected D-arginine even in the absence of etoposide in HepG2 cells. However, unexpectedly, the concentrations of all 15 D-amino acids were below the limit of quantification. We suppose this result may raise two possibilities of the regulation of D-amino acids production in U2OS cells. The first possibility is that the D-amino acid(s) required for DAO-mediated senescence promotion may be other types of D-amino acids. Since DAO is known to oxidize a broad range of D-amino acids including D-norleucine and D-ornithine, we consider that DAO can also promote senescence through the oxidation of D-amino acids other than those tested in our laboratory. So far, unfortunately, we have been unable to measure the levels of other D-amino acids such as D-norleucine and D-ornithine owing to the unavailability of the experimental system for detection and quantification of such D-amino acids. However, regardless of whether or not D-norleucine and D-ornithine are the physiological substrates in U2OS cells, we consider that D-serine and D-arginine can function as DAO substrates because the external addition of these D-amino acids to the medium potentiated the DAO effect on

senescence promotion (Fig. 4). The second possibility is that the D-amino acid(s) required for DAO-mediated senescence promotion may be temporally produced during senescence by amino acid racemases that convert L-amino acids to D-amino acids. We think that these are very important and interesting questions for understanding the regulation of DAO-mediated senescence promotion. Therefore, we intend to clarify these issues in future. In addition, to more clearly explain the point raised by the reviewer, we have added the following sentence to the Discussion section, “Actually, so far we have been unable to detect D-arginine and D-serine at quantifiable levels in normally proliferating U2OS cells, suggesting that the D-amino acid(s) required for DAO-mediated senescence promotion varies depending on the cell type and/or on the senescence stage.” (P. 18, lines 14-17).

(6) In Fig4A, SA-β-gal positive cell was increased when only treated with Riboflavin and D-serine (compare bars 1 and 3, 4 and 6). Could DNA damage be also detected in these cells?

We agree that it would be helpful to determine whether the combined treatment with riboflavin and D-serine induces DNA damage. Therefore, we have analyzed the protein level of γ -H2AX, a DNA damage marker, and found that the γ -H2AX level was slightly increased by the treatment with riboflavin and D-serine, as shown in Figure 3 in this rebuttal letter (please see below). Although this result is fundamentally consistent with the conclusion obtained from SA- β -gal and EdU incorporation assays, we suspect that this is too preliminary to report in detail as yet, as can be seen from the figure. We are now investigating whether and how riboflavin and D-serine are related to DNA damage and intend to report it in a later publication.

Figure 3. U2OS cells were treated with 50 μ M riboflavin and 5 mM D-serine as indicated. After incubation for 7 days, the cells were subjected to immunoblot analysis with anti- γ -H2AX antibody (Merck Millipore; 05-636). The protein levels relative to the γ -tubulin levels were quantified using NIH ImageJ software and are indicated at the bottom of each lane. The protein level of γ -H2AX, a DNA damage marker, was slightly increased by the combined treatment with riboflavin and D-serine.

All changes made:

Main Figures

The representative images in Figures 1D, 2C and D, and 3C were omitted.

The relative protein levels were indicated at the bottom of each lane of the blots in Figures 1F and 2E and F

The old Figure 5C-E concerning the cytosolic-localized mutant of DAO were omitted; accordingly, the related text in the Results and Materials and Methods sections were removed.

Several new panels were added:

The new Figure 1F shows that etoposide-induced upregulation of p21 is impaired by knockdown of DAO.

The new Figure 2I shows that etoposide-induced upregulation of p21 is impaired by DAO inhibition in Hs68 cells.

The new Figure 2J-L show that oncogenic Ras-induced senescence is impaired by DAO inhibition in WI-38 cells.

The new Figure 5A and B show that intracellular FAD concentration is increased in response to etoposide treatment, which is abrogated by knockdown of RFVT-1.

The new Figure 6G shows that DAO-promoted upregulation of p21 is impaired by ROS scavenging.

The new Figure 6J shows that DAO inhibition had no synergistic effect on p21 repression when combined with ROS scavenging.

The new Figure 6M and P show that DAO and PRODH cooperatively contribute to p21 upregulation in U2OS and Hs68 cells.

The text and figure legends were altered accordingly.

October 10, 2018

Re: Life Science Alliance manuscript #LSA-2018-00045-TR

Prof. Shinji Kamada
Kobe University
Biosignal Research Center
1-1 Rokkodai-cho, Nada-ku
Kobe, Hyogo 657-8501
Japan

Dear Dr. Kamada,

Thank you for submitting your revised manuscript entitled "D-amino acid oxidase promotes cellular senescence via the production of reactive oxygen species" to Life Science Alliance. The manuscript was assessed by the original expert reviewers again, whose comments are appended to this letter.

As you will see, reviewer#2 appreciates the changes introduced during revision and now supports publication. Reviewer #1 and #3, however, do not support publication at this stage and think that their initial concerns are not sufficiently addressed.

We think that the major concern put forward by both reviewers can get addressed by providing better data for the p21 analysis (more replicates and statistics), and we would be happy to publish such a further revised version in Life Science Alliance. When further revising your work, please also normalize the phospho-p53 levels to total p53 levels (reviewer #1).

The typical timeframe for revisions is three months.

Thank you for this interesting contribution to Life Science Alliance. We are looking forward to receiving your revised manuscript.

Sincerely,

- A letter addressing the reviewers' comments point by point.
- An editable version of the final text (.DOC or .DOCX) is needed for copyediting (no PDFs).
- High-resolution figure, supplementary figure and video files uploaded as individual files: See our detailed guidelines for preparing your production-ready images, <http://life-science-alliance.org/authorguide>
- Summary blurb (enter in submission system): A short text summarizing in a single sentence the study (max. 200 characters including spaces). This text is used in conjunction with the titles of papers, hence should be informative and complementary to the title and running title. It should describe the context and significance of the findings for a general readership; it should be written in the present tense and refer to the work in the third person. Author names should not be mentioned.

B. MANUSCRIPT ORGANIZATION AND FORMATTING:

Full guidelines are available on our Instructions for Authors page, <http://life-science-alliance.org/authorguide>

Reviewer #1 (Comments to the Authors (Required)):

In their revised manuscript, Nagano et al. have addressed several of the reviewers' concerns. 1) To provide additional characterization of the senescence phenotype, they include analysis of p21 levels in several of their experiments. 2) They now report that oncogene-induced senescence also involves DAO, showing that their findings are not limited to DNA damage-induced senescence. 3)

The authors have depleted DAO using RNAi, thus bolstering their previous results that were based on use of a DAO chemical inhibitor. 4) They show that the FAD transporter, RFVT1, is induced by etoposide treatment and its ablation decreases cellular FAD levels. These data suggest that increased FAD, a DAO cofactor, may enhance DAO activity in senescence cells.

These new data have improved the manuscript. It should be noted that p21 is not a new senescence marker in their study since this protein was analyzed to a lesser extent in the previous version of the paper. Also, the effects seen on p21 levels tend to be minor and the quantitation appears to be from a single measurement with no statistical analysis included. It is unfortunate that the authors did not examine expression of SASP genes, which represent independent markers of senescence.

The quantitation of p53-pS15 in Fig 2E-F should be normalized to total p53 levels, as this is the correct way to measure changes in specific phosphorylation of the protein.

On page 11. The sentence "...knockdown of DAO suppressed the etoposide-induced ROS accumulation...." should be changed to "partially suppressed," which is a more accurate description.

Reviewer #2 (Comments to the Authors (Required)):

I think that the revised manuscript has substantially been improved and now is acceptable for publication.

Reviewer #3 (Comments to the Authors (Required)):

The authors have now resubmitted a revised version of the manuscript considering concerns raised by the reviewers. The new version of the manuscript is improved, and the conclusions are better supported. However, I still having a major concern that precludes me to accept the manuscript in the current version. Although the authors added the p21 data in Figure 1, 2 and 6, the quantitative data of DNA damage signaling are needed as I suggested before. Because the authors state that DAO promotes DNA damage induced senescence. However, there is no data showing the DNA damage in all figures. I therefore feel that this manuscript is rather preliminary and would not be of sufficient general interest for the readership of Life Science Alliance.

Reviewer #1 (Comments to the Authors (Required)):

In their revised manuscript, Nagano et al. have addressed several of the reviewers' concerns. 1) To provide additional characterization of the senescence phenotype, they include analysis of p21 levels in several of their experiments. 2) They now report that oncogene-induced senescence also involves DAO, showing that their findings are not limited to DNA damage-induced senescence. 3) The authors have depleted DAO using RNAi, thus bolstering their previous results that were based on use of a DAO chemical inhibitor. 4) They show that the FAD transporter, RFVT1, is induced by etoposide treatment and its ablation decreases cellular FAD levels. These data suggest that increased FAD, a DAO cofactor, may enhance DAO activity in senescence cells.

These new data have improved the manuscript. It should be noted that p21 is not a new senescence marker in their study since this protein was analyzed to a lesser extent in the previous version of the paper. Also, the effects seen on p21 levels tend to be minor and the quantitation appears to be from a single measurement with no statistical analysis included. It is unfortunate that the authors did not examine expression of SASP genes, which represent independent markers of senescence.

We agree with the reviewer that it is important to examine the expression of SASP factors as an independent marker of senescence. Therefore, we have analyzed the expression level of IL-6, a key SASP factor, and the results show that the expression level of IL-6 was upregulated by the combined treatment with riboflavin and D-serine in DAO-overexpressing cells to a level comparable with that in etoposide-induced senescent cells. This result supports our conclusion that riboflavin and D-serine enhanced the DAO effect on the induction of senescence. These new data have been added as Figure 5E, and the manuscript has been modified accordingly.

The quantitation of p53-pS15 in Fig 2E-F should be normalized to total p53 levels, as this is the correct way to measure changes in specific phosphorylation of the protein.

According to the reviewer's suggestion, the phosphorylated p53 (p53 pS15) levels have been re-quantified and now shown as levels relative to total p53, and the Figure legends of Figure 2E and F have been modified accordingly.

On page 11. The sentence "...knockdown of DAO suppressed the etoposide-induced ROS accumulation...." should be changed to "partially suppressed," which is a more accurate description.

As instructed by the reviewer, we have revised the text.

Reviewer #2 (Comments to the Authors (Required)):

I think that the revised manuscript has substantially been improved and now is acceptable for publication.

Reviewer #3 (Comments to the Authors (Required)):

The authors have now resubmitted a revised version of the manuscript considering concerns raised by the reviewers. The new version of the manuscript is improved, and the conclusions are better supported. However, I still having a major concern that precludes me to accept the manuscript in the current version. Although the authors added the p21 data in Figure 1, 2 and 6, the quantitative data of DNA damage signaling are needed as I suggested before. Because the authors state that DAO promotes DNA damage induced senescence. However, there is no data showing the DNA damage in all figures. I therefore feel that this manuscript is rather preliminary and would not be of sufficient general interest for the readership of Life Science Alliance.

We agree with the reviewer that it is important to test whether DNA damage is induced during DAO-mediated senescence. Therefore, we have carried out the experiment to obtain quantitative data of DNA damage signaling, and the results show that the combined treatment with riboflavin and D-serine induced DNA damage in DAO-overexpressing cells. This result supports our conclusion that DAO promotes DNA damage-induced senescence under the condition where riboflavin (the precursor of coenzyme) and D-serine (substrate of DAO) are abundantly present. These new data have been added as Figure 5F, and the manuscript has been modified accordingly.

All changes made:

Main Figures

The relative protein levels of phosphorylated p53 (p53 pS15) were normalized to the total p53 levels and indicated at the bottom of each lane of the blots in Figures 2E and F.

Two new panels were added:

The new Figure 5E shows that IL-6 was upregulated by the combined treatment with riboflavin and D-serine in DAO-overexpressing cells.

The new Figure 5F shows that DNA damage was induced by the combined treatment with riboflavin and D-serine in DAO-overexpressing cells.

The text and figure legends were altered accordingly.

January 11, 2019

RE: Life Science Alliance Manuscript #LSA-2018-00045-TRR

Prof. Shinji Kamada
Kobe University
Biosignal Research Center
1-1 Rokkodai-cho
Nada-ku
Kobe, Hyogo 657-8501
Japan

Dear Dr. Kamada,

Thank you for submitting your revised manuscript entitled "D-amino acid oxidase promotes cellular senescence via the production of reactive oxygen species". We appreciate that this revised version now additional senescence marker analysis and DNA damage analysis, and we are thus happy to publish your paper in Life Science Alliance.

Before sending you the official acceptance letter, please log into our system one more time to fill in the electronic license to publish form. Your manuscript number will change to LSA-2018-00045-TRRR when doing so, please make sure to move all manuscript files to this new manuscript version (single click process).

Please log in to your account: <https://lsa.msubmit.net/cgi-bin/main.plex>
You will be guided to complete the submission of your revised manuscript and to fill in all necessary information.

A. FINAL FILES:

-- High-resolution figure, supplementary figure and video files uploaded as individual files: See our detailed guidelines for preparing your production-ready images, <http://life-science-alliance.org/authorguide>

B. MANUSCRIPT ORGANIZATION AND FORMATTING:

Full guidelines are available on our Instructions for Authors page, <http://life-science-alliance.org/authorguide>

Sincerely,

January 11, 2019

RE: Life Science Alliance Manuscript #LSA-2018-00045-TRRR

Prof. Shinji Kamada
Kobe University
Biosignal Research Center
1-1 Rokkodai-cho
Nada-ku
Kobe, Hyogo 657-8501
Japan

Dear Dr. Kamada,

Thank you for submitting your Research Article entitled "D-amino acid oxidase promotes cellular senescence via the production of reactive oxygen species". It is a pleasure to let you know that your manuscript is now accepted for publication in Life Science Alliance. Congratulations on this interesting work.

*****IMPORTANT:** If you will be unreachable at any time, please provide us with the email address of an alternate author. Failure to respond to routine queries may lead to unavoidable delays in publication.*******

DISTRIBUTION OF MATERIALS:

Again, congratulations on a very nice paper. I hope you found the review process to be constructive and are pleased with how the manuscript was handled editorially. We look forward to future exciting

submissions from your lab.

Sincerely,
